# Deciphering the molecular logic of WOX5 function in the root stem cell organizer

Ning Zhang [ID] [1,2,4 ✉], Pamela Bitterli [ID] [2,4], Peter Oluoch [ID] [2], Marita Hermann[2], Ernst Aichinger[2], Edwin P Groot[2,3] & Thomas Laux [ID] [2,3 ✉]

## Abstract

Plant and animal stem cells receive signals from their surrounding cells to stay undifferentiated. In the Arabidopsis root, the quiescent center (QC) acts as a stem cell organizer, signaling to the neighboring stem cells. WOX5 is a central transcription factor regulating QC function. However, due to the scarcity of QC cells, WOX5 functions in the QC are largely unexplored at a genomic scale. Here, we unveil the transcriptional and epigenetic landscapes of the QC and the role of WOX5 within them. We find that WOX5 functions both as a transcriptional repressor and activator, affecting histone modifications and chromatin accessibility. Our data expand on known WOX5 functions, such as the regulation of differentiation, cell division, and auxin biosynthesis. We also uncover unexpected WOX5-regulated pathways involved in nitrate transport and the regulation of basal expression levels of genes associated with mature root tissues. These data suggest a role for QC cells as reserve stem cells and primed cells for prospective progenitor fates. Taken together, these findings offer insights into the role of WOX5 at the QC and provide a basis for further analyses to advance our understanding of the nature of plant stem cell organizers.

**Keywords** WOX5; Stem Cells Organizer; Cell-specific Transcriptomic and Epigenetic Profiling; Root; Arabidopsis
**Subject Categories** Chromatin, Transcription & Genomics; Plant Biology

## Introduction

Stem cells in plants and animals are situated in specific environments, named stem cell niches, which provide signals to keep the stem cells undifferentiated. The niche cells in animals are often located at one side of a polar stem cell system, acting as a source of a stem cell-maintaining signal gradient. When stem cell daughter cells leave the proximity of the niche cells, the waning signal strength eventually allows for entry into differentiation. Plant cells are immobile and tightly embedded into a grid of cell walls. Their niche cells are organized in small cell groups, the stem cell organizers, and consist of changing constituent cells in root and shoot meristems (Laux, 2003).

The stem cells that give rise to the different files of cell types in the root are localized around the QC cells (Dolan et al, 1993; Pardal and Heidstra, 2021; Rahni and Birnbaum, 2019): The proximal stem cells give rise to the stele, endodermis, and cortex, the lateral ones to the epidermis and the lateral root cap, and the distal stem cells to the gravity-sensing columella (Dolan et al, 1993). The QC was initially characterized in the 1950s by Clowes as a region of low mitotic activity (Clowes, 1953, 1956). Subsequent research has clarified the role of the QC as a stem cell organizer: laser ablation of the QC cells resulted in the neighboring stem cells ceasing division and initiating differentiation (van den Berg et al, 1997). Recent studies revealed, however, that the QC is dynamic and fulfills additional roles. For example, the *Arabidopsis* QC is mitotically inactive at the young seedling stage but gradually becomes mitotically active over time (Timilsina et al, 2019; Wein et al, 2020), resulting in an open (dividing QC) root meristem structure typical for plant species with larger roots, such as the common sunflower (Clowes, 1981). Second, elegant ablation experiments revealed that the QC can take over stem cell function after injury (Zhou et al, 2019). This has led to the hypothesis that the QC cells can also act as a reservoir to replenish stem cells when necessary.

Unlike the stem cells for all other root cell files, the asymmetric divisions of the CSCs give rise to daughter cells that do not partake in further divisions but undergo direct differentiation into columella cells (CCs) (Dolan et al, 1993). The differentiated CCs accumulate starch grain-containing amyloplasts, the critical organelles of gravity sensing. Because of its simplicity, the *Arabidopsis* columella stem cells niche, consisting of the QC, one layer of CSCs, and several layers of differentiated CCs, has become a model system to study stem cell regulation.

The *WUSCHEL RELATED HOMEOBOX 5* (*WOX5*), which is specifically expressed in the QC cells (Haecker et al, 2004; Sarkar et al, 2007) regulates several important features of the QC. Firstly, WOX5 promotes the mitotic quiescence of the QC by direct transcriptional repression of the D-type *CYCLIN 3;3* (*CYCD3;3*) (Forzani et al, 2014). This activity involves direct interaction with the transcription factors BRASSINOSTEROIDS AT VASCULAR

[1]State Key Laboratory of Wheat Improvement, College of Agronomy, Shandong Agricultural University, 271018 Tai'an, Shandong, China. [2]Signalling Research Centres BIOSS and CIBSS, Faculty of Biology, University of Freiburg, Schänzlestrasse 1, 79104 Freiburg, Germany. [3]Sino-German Joint Research Center on Agricultural Biology, Shandong Agricultural University, Tai'an, Shandong, China. [4]These authors contributed equally: Ning Zhang, Pamela Bitterli. ✉E-mail: nzhang@sdau.edu.cn; laux@biologie.uni-freiburg.de

AND ORGANIZING CENTER (BRAVO) and PLETHORA3 (PLT3) (Betegon-Putze et al, 2021; Burkart et al, 2022). Secondly, it maintains the QC cells undifferentiated, in part by directly repressing the differentiation gene *CDF4* (Pi et al, 2015; Sarkar et al, 2007). Thirdly, WOX5 expression in the QC is non-cell-autonomously necessary to maintain the underlying columella stem cells (CSCs) undifferentiated (Sarkar et al, 2007). Finally, WOX5 promotes local biosynthesis of the phytohormone auxin in the QC (Savina et al, 2020; Sharma et al, 2024), which, together with shoot-derived auxin, determines an auxin response maximum that is essential for QC function (Gälweiler et al, 1998; Grieneisen et al, 2007). On the other side, WOX5 expression is also positively regulated by auxin, suggesting a feed-forward loop between WOX5 and auxin in QC cells (Ding and Friml, 2010; Gonzali et al, 2005; Sarkar et al, 2007). The current view is, therefore, that auxin and WOX5 provide coordinated and parallel inputs into QC function (Bennett et al, 2014; Sharma et al, 2024). Furthermore, auxin biosynthesis in the QC affects the position of the transition border where meristem cells rapidly elongate, suggesting a long-range signal from the QC that acts over more than ten cells (Moubayidin et al, 2013). In addition to these WOX5-regulated features, the expression of several QC-specific enhancer trap lines is either completely abolished or deregulated in the *wox5-1* mutant (Sarkar et al, 2007). However, no gene functions have yet been associated with these enhancer trap lines. A recent study of the *Arabidopsis* root stem cell niche revealed stem cell-specific gene networks (Clark et al, 2019), providing a resource for future studies. Complementary to the termination of CSCs in the loss-of-function mutant, ectopic expression of WOX5 is sufficient to reprogram differentiated columella cells into CSC-like cells (Pi et al, 2015; Sarkar et al, 2007), providing an experimental tool for molecular studies requiring larger amounts of cells.

The WOX5 protein sequence contains two conserved elements, the WUS box (WB) and an ethylene-responsive element binding factor-associated amphiphilic repression (EAR) domain, both essential for interaction with TOPLESS (TPL)/TOPLESS RELATED (TRP) co-repressors (Daum et al, 2014; Hiratsu et al, 2003; Ikeda et al, 2009; Kieffer et al, 2006; Pi et al, 2015). To down-regulate the expression of the differentiation factor *CYCLIC DOF FACTOR 4* (*CDF4*) in the QC, WOX5 recruits the TPL/TPR/HISTONE DEACETYLASE 19 (HDA19) complex to the *CDF4* promoter, reducing the H3K9ac and H3K14ac marks that are associated with transcriptionally active chromatin. The WOX5-mediated decreased *CDF4* transcript levels in the QC and the CSCs generate opposing WOX5/CDF4 gradients to balance stem cell maintenance and differentiation (Pi et al, 2015). However, *CDF4* expression in the QC from the *WOX5* promoter causes less starch accumulation at the CSC position than observed in the *wox5-1* mutant, suggesting that while WOX5-mediated repression of *CDF4* is essential to maintain the CSC undifferentiated, additional pathways regulated by WOX5 must exist (Berckmans et al, 2020; Pi et al, 2015).

Despite its multiple functions, the nature of the QC as a stem cell organizer and how WOX5 controls the functions are still largely unknown. This is partly due to the limited quantity of QC cells in a root, which has posed a challenge to analyzing QC cells at the molecular level. Through nuclei sorting, we provide a comprehensive picture of the transcriptional and epigenetic architecture of QC cells and the role of WOX5 therein. Our data confirm and expand the roles of WOX5 in repressing differentiation and cell division and promoting auxin biosynthesis in the QC. In addition, we provide evidence for WOX5 function in regulating unexpected pathways, suggesting, for example, that WOX5 might be involved in priming the QC towards root cell fates.

# Results

## Nuclear transcriptomes reveal QC- and CC-specific programs

To gain insight into the molecular nature of the QC cells, we initially sought to purify QC protoplasts by Fluorescent-Activated Cell Sorting (FACS). However, our initial experiments did not yield robust results, possibly due to inevitable cellular stress during the required enzymatic cell wall degradation of the deeply embedded QC cells and/or the previously observed ectopic activation of WOX5 expression during this procedure (Birnbaum et al, 2003; Denyer et al, 2019). Since the nuclear transcriptome is plausibly a more direct readout of WOX5 activity, we, therefore, focused on Fluorescent-Activated Nuclei Sorting (FANS, Fig. 1A) (Slane et al, 2014) from QC cells, marked by *pWOX5:H2B-tdTomato*, and CCs as a reference, marked by *PET111::H2B-tdTomato*, in 5-day-old wild-type seedlings (Fig. 1B,C; Dataset EV1). All obtained low-input genomic transcriptome, histone modification, and chromatin accessibility data were of good quality (Fig. 1D–G; Appendix Fig. S1), confirming the suitability of this approach.

We identified 1709 differentially expressed genes (DEGs) expressed at a higher level in the QC nuclei (hereafter: QC-up DEGs) than in the CC nuclei. On the other hand, 2168 genes were expressed at higher levels in the CCs (hereafter: CC-up DEGs) than in QC ($P$adj <0.05, FC ± 1.5) (Fig. 2A; Dataset EV2). We confirmed the differential expression for 16 out of 20 randomly selected genes by RT-qPCR from independently sorted nuclei (Table EV1). Furthermore, we found that the transcript levels of most published QC-specifically expressed genes were between 2 and 43.5 times higher in our QC than in CC data (Fig. EV1A; Table EV2). Also, the levels of several published CC-enriched transcripts were between 2.7 and 26.4 times higher in our CC compared to the QC data (Fig. EV1A; Table EV2). Thus, we conclude that the obtained data faithfully represent the nuclear QC and CC transcriptomes.

To gain insight into to which extent nuclear transcripts can predict cellular transcriptomes of QC and CCs, we compared our nuclear transcriptome data to published transcriptomes obtained after fluorescent-activated sorting of protoplasts (Li et al, 2016). To this end, we defined DEGs from the protoplast data by the same criteria as our FANS-derived data ($P$adj < 0.05, FC ± 1.5). Initially, we observed a poor overlap between the two data sets. However, after removing one outlier replicate of the protoplast data (Fig. EV1B), the two data sets showed a high Spearman correlation (Fig. EV1C) and an overlap of 46.87% (801/1709) for the QC-up and 53.41% (1158/2168) for the CC-up (Fig. EV1D). The shared DEGs include the overrepresented gene ontology (GO) terms "cell cycle" and "meristem maintenance" in the QC-up and "cell wall biogenesis" and "root cap development" in the CC-up genes (Fig. EV1E), representing known functions of QC and CCs. On the other hand, the DEGs identified from protoplasts but not in our FANS data were enriched for GO categories related to stress and

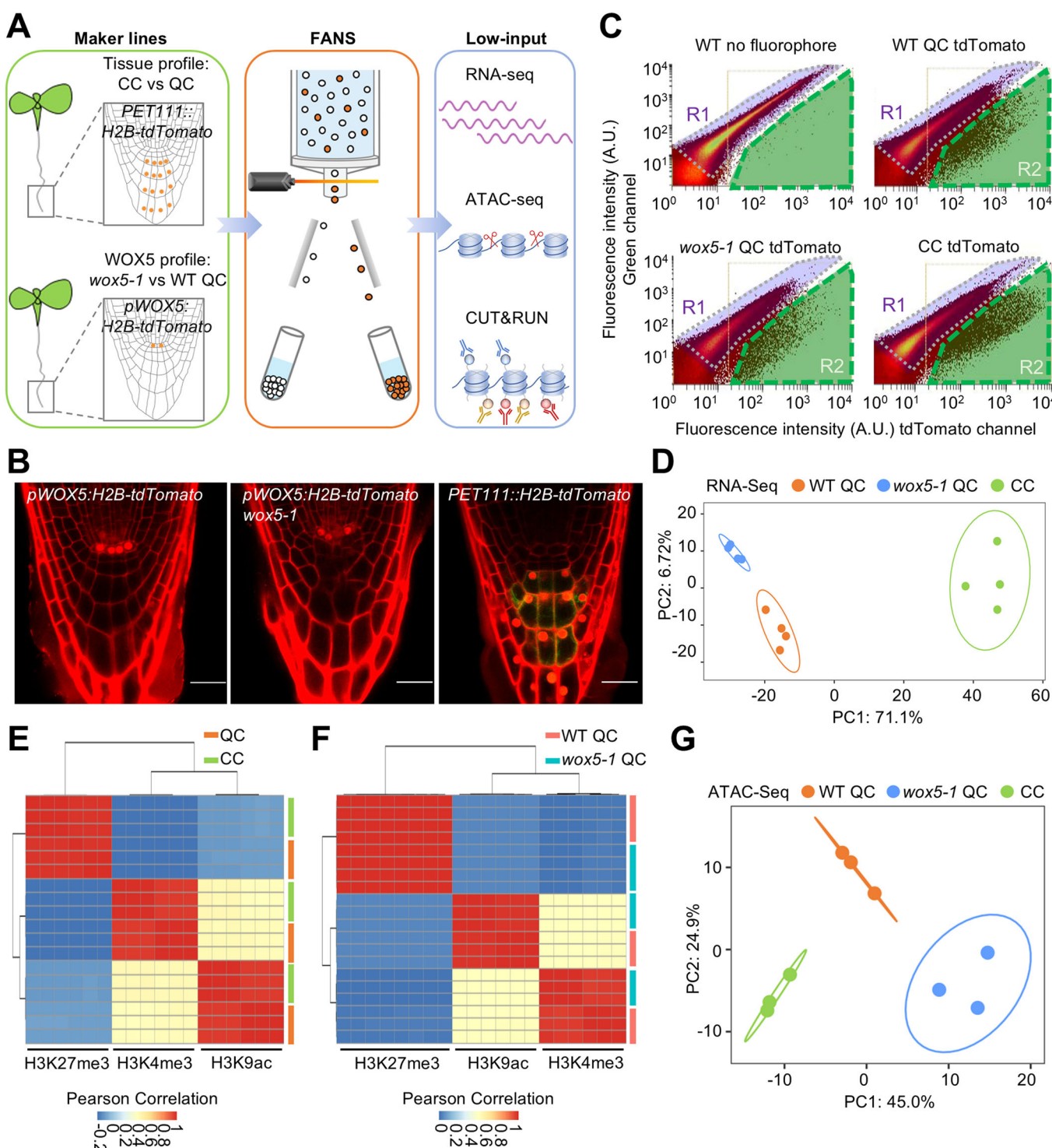

mechanical stimulus, possibly reflecting cellular stress during protoplast isolation (Fig. EV1E).

We next asked whether our data share similarities with publicly available transcriptomes of cell types outside the columella stem cell niche. We found that our QC-up nuclear DEGs are generally associated with published transcriptomes from immature cell types (Li et al, 2016; Shi et al, 2021; Yadav et al, 2014), such as the root

meristematic zone or the L2 and L3 layers, and abaxial and adaxial organ boundaries of the shoot meristem (Fig. EV2A–C). Conversely, CC-up nuclear DEGs overlapped with published transcriptomes from more mature cell types, such as root maturation and elongation zones or the starch sheath in inflorescences (Fig. EV2A–C). Thus, nuclear QC and CC transcriptomes share expression profiles with cell types at similar maturation stages.

**Figure 1. Low-input analysis through FANS.**

(A) Schematic representation of the experimental design. Nuclei of wild-type QC cells and CCs and *wox5-1* QC cells were collected by FANS to perform genome-wide low-input RNA-Seq, ATAC-Seq, and histone CUT&RUN analysis. (B) QC and CC-specific marker lines used for FANS. Bars = 20 µm. (C) FANS plot derived from the indicated genotypes. The nuclei were sorted by the strength of the tdTomato signal in the R2 region (green). Wild-type plants without fluorophore were used to set the R1 region (purple) as a negative control. The y axis indicates the autofluorescence intensity, and the x axis indicates the tdTomato fluorescence intensity. (D) PCA of the indicated FANS RNA-Seq data. Each dot represents one biological replicate. The first two dimensions with the largest effects are shown. (E, F) Heatmap showing the Pearson correlation of the biological replicates of the indicated histone modification CUT&RUN experiments for CC and QC nuclei (E), and *wox5-1* QC and wild-type QC nuclei (F). (G) PCA of the indicated ATAC-Seq data. Each dot represents one biological replicate. The first two dimensions with the largest contributions are shown. Source data are available online for this figure.

To decipher which cellular functions are differently encoded between the nuclear CC and QC transcriptomes, we performed a GO analysis. We found that in the QC-up DEGs, the over-represented GO terms include "chromatin modifications," "cell cycle," and "meristem maintenance" (Fig. 2B; Dataset EV3). The higher expression of cell cycle genes in the "quiescent" QC cells might be surprising but is consistent with the low mitotic activity of the QC compared to the non-dividing CCs at the seedling stage (Betegon-Putze et al, 2021; Ornelas-Ayala et al, 2020; Rahni and Birnbaum, 2019; Savina et al, 2020; Zhang et al, 2013). We also identified enriched GO biological processes in QC cells that were not previously linked with root stem cells, such as several genes encoding nitrogen metabolism enzymes (Fig. 2B; Dataset EV3). For instance, the *CYSTATHIONINE BETA-LYASE* (*CBL*) gene encodes the second enzyme in the methionine biosynthetic pathway (Liu et al, 2019). Notably, *CBL* loss-of-function results in defective QC maintenance and root development (Liu et al, 2019), suggesting the functional relevance of its upregulation in the QC.

On the other hand, CC-up DEGs overrepresent GO terms, including "root cap development," "epidermal differentiation," and "programmed cell death" (Fig. 2C; Dataset EV3), consistent with the differentiated nature of the CCs.

## The QC- and CC-specific nuclear transcriptomes are associated with different chromatin architectures

We then asked whether nuclear CC and QC transcriptional profiles are associated with specific chromatin landscapes. To this end, we analyzed histone modifications by CUT&RUN experiments (Fig. 1A,E; Appendix Fig. S1A) and chromatin accessibility by ATAC-Seq with sorted nuclei (Fig. 1A,G; Appendix Fig. S1C). We chose H3K9ac and H3K4me3 as marks associated with transcriptionally competent chromatin because previous studies showed that WOX5 directly modifies H3K9ac in the *CDF4* gene (Pi et al, 2015) and, together with H3K4me3, is often found near the TSSs where also WOX5 binds. As a mark for repressive chromatin, we chose H3K27me3, a hallmark of transcriptional downregulation in plants and animals (Wiles and Selker, 2017).

We defined differential histone modifications (dHMs) and differentially accessible chromatin regions (dACRs) between CC and QC nuclei when corresponding peak values differed with a *P*adj <0.05. We then assigned a dHM or a dACR to a gene when it was located between 3 kb upstream of the transcriptional start site (TSS) and 0.5 kb downstream of the transcriptional termination site (TTS) (Dataset EV4). This covered the majority of dHMs and dACRs between CC and QC nuclei, with only less than 7% of each located in the intergenic regions (Fig. 2D, Appendix Fig. S2A–D). Moreover, the transcriptome differences between CC and QCs correlated with corresponding dHMs and dACRs differences (Fig. 2E,F).

"Bivalent" chromatin domains, characterized by overlapping activating (e.g., H3K4me3 and/or H3K9ac) and repressive (e.g., H3K27me3) histone modifications, have been associated with a poised state of gene transcription in stem cells (Blanco et al, 2020). Interestingly, we observed a slightly higher frequency of genes with bivalent chromatin domains in the QC cells than in the CCs in the upstream 3 kb of TSS (Appendix Fig. S2E; Dataset EV5).

Thus, the cell type-specific transcriptomes of CC and QC cells are highly associated with altered histone modifications and chromatin accessibility.

## WOX5-regulated genes encode key pathways of the QC

We then addressed the role of WOX5 in regulating the transcriptional architecture of the QC. To identify WOX5-regulated genes in the QC, we compared the transcriptomes of sorted nuclei expressing *pWOX5:H2B-tdTomato* from *wox5-1* and wild-type roots (Fig. 1A–D; Dataset EV1). We found that 1423 genes were positively ("WOX5-activated", wild-type QC > *wox5-1* QC) and 947 genes were negatively ("WOX5-repressed", wild-type QC < *wox5-1* QC) associated with WOX5 activity (*P*adj <0.05; Fig. 3A; Dataset EV6). We validated the RNA-Seq data for 17 out of 20 randomly selected genes by RT-qPCR (Table EV1). Furthermore, our WOX5 DEGs significantly overlapped with data obtained from bulk RNA-Seq of entire wild-type and *wox5-1* root tips (Clark et al, 2020) (Fig. EV3A). The latter study also identified many candidate WOX5-regulated genes not found in our study. However, many of those transcripts had previously been reported to be enriched in root cell types outside the QC, such as meristematic zone, mature zone, columella, and hair cells (Fig. EV3B). We noted that the previously identified WOX5 downstream genes *HAN*, *TAA1*, *CDF4*, and *CYCD3;3* (Forzani et al, 2014; Pi et al, 2015; Savina et al, 2020; Sharma et al, 2024) displayed the expected regulation in our genomic data but did not meet our criteria for statistical significance due to their variance between the genomic replicates in our transcriptome profiling (Table EV3). Therefore, we confirmed the expected WOX5-dependent expression in cDNA pools from separately sorted nuclei for all of these genes by RT-qPCR (Appendix Fig. S3). In summary, these validations suggest that data from sorted nuclei faithfully represent the WOX5-regulated DEGs in the QC.

Importantly, the WOX5-regulated DEGs confirm and expand on known functions of WOX5 in the QC: repression of cell division (*CYCD3.3*, *KRP1*, *KRP2*, *SIM*, *BORI2*) and differentiation (*CDF4*, *BBM*, *RGF8*, *HB-8*, *QQS*), and promoting auxin signaling (*HAN*, *TAA1*) (Table EV4). On the other hand, comparing the WOX5-regulated genes with our nuclear QC and CC transcriptome data, we found only limited, though statistically significant, overlap (Appendix Fig. S4), including the upregulation by WOX5 of the

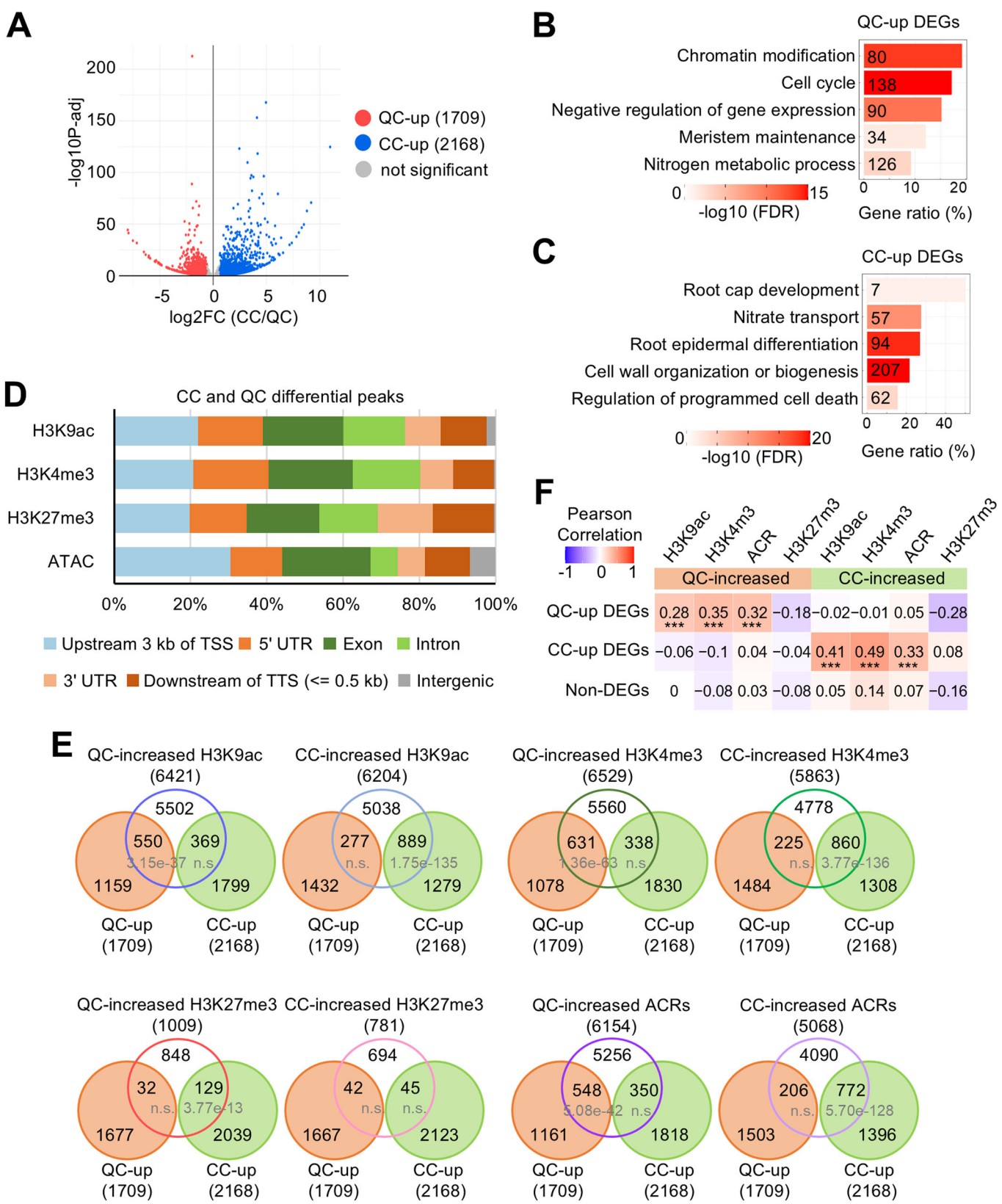

**Figure 2. Expression of CC and QC DEGs is associated with dHMs and dACRs in CC and QC cells.**

(A) Volcano plot of the DEGs between CC and QC from four biological replicates (FC ± 1.5, Padj <0.05). Padj: Benjamini–Hochberg adjusted $P$ value. (B, C) GO analysis of QC-up DEGs (B) and CC-up DEGs (C). $X$ axis shows the proportion of genes per GO term (Gene ratio). The color gradient indicates –log10 (FDR) by Fisher exact test and Yekutieli correction (FDR < 0.05). The number of genes in each term is shown. (D) Genomic distributions of dHMs and dACRs over the indicated regions. (E) Venn diagrams of QC-up/CC-up DEGs, dHM- and dACR-assigned genes, respectively. $P$ values indicate statistical significance by hypergeometric tests. QC-up, QC-up DEGs. CC-up, CC-up DEGs. (F) Pearson correlation of QC-up, CC-up and non-DEGs (not differentially expressed between CC and QC), dHMs (CC/QC), and dACRs (CC/QC). The colors and numbers represent the Pearson correlation coefficients. Correlation $P$ values between QC-up DEGs and QC-increased H3K9ac, H3K4me3, and ACR are 1.15e-11, 3.72e-20, and 3.39e-14, respectively. Correlation $P$ values between CC-up DEGs and CC-increased H3K9ac, H3K4me3, and ACR are 9.19e-37, 1.72e-52, and 1.33e-20, respectively. Source data are available online for this figure.

published QC-specifically expressed genes *BABY BOOM* (*BBM*) and *ROOT MERISTEM GROWTH FACTOR 8* (*RGF8*) (Aida et al, 2004; Denyer et al, 2019; Fernandez et al, 2013), and the starch accumulation repressor gene *Qua-Quine Starch* (*QQS*) (Li et al, 2009), consistent with the absence of starch in the QC (Table EV4). Thus, WOX5 regulates multiple aspects of the key features of the QC, mitotic quiescence, differentiation state, and auxin signaling, rather than global transcriptome differences between QC and CC.

## WOX5 acts mainly as a transcriptional activator affecting the chromatin state of putative direct target genes

To search for potential direct targets of WOX5 in the QC, we isolated WOX5-binding genomic DNA fragments with anti-WOX5 antibodies. Due to the limited number of QC cells in a root meristem, Chromatin Immunoprecipitation (ChIP) or CUT&RUN (Skene and Henikoff, 2017) experiments did not yield robust results from sorted QC nuclei. Therefore, we decided to use the previously characterized *p35S:WOX5-GR* overexpression line (Pi et al, 2015), which accumulates undifferentiated cells in place of differentiated CCs, as a proxy for WOX5 function, comparing it to the knockout mutant *wox5-1* as a negative control (Appendix Fig. S5). Of the WOX5-bound-regions (WBR) identified by this approach (FC > 2 and Padj <0.05; Dataset EV7), 93% were located within upstream 3 kb of the TSS and downstream 0.5 kb of the TTS of annotated genes, and only 7% in intergenic regions (Fig. 3B). The vast majority of these genes were bound by WOX5 at one or multiple sites within 3 kb upstream of the TSS (Fig. 3B), which encompasses the region typically bound by plant transcription factors (Cheng et al, 2023; Shahmuradov et al, 2017; Yu et al, 2016). To discriminate against false positive WBR, which are notorious for overexpressed transcription factors, we selected the WOX5-bound genes that WOX5 differentially regulated in the QC (Fig. 3C). Therefore, we focused on these 812 upregulated and 259 down-regulated DEGs, which are bound and regulated by WOX5 as the putative direct WOX5 targets for further analysis (Fig. 3C). We confirmed the two published directly repressed WOX5 target genes *CYCD3.3* and *CDF4* among the potential directly downregulated WOX5 targets (Dataset EV7; Table EV5).

However, our results suggested that WOX5, initially only characterized as a transcriptional repressor (Forzani et al, 2014; Pi et al, 2015), acts in transcriptional activation of most of its target genes. Since WOX5 has been reported to repress its target gene *CDF4* by recruiting TPL/HDA19 and removing H3K9ac marks (Pi et al, 2015), we hypothesized that alternative mechanisms must be in place for directly activated WOX5 DEGs. Therefore, we investigated the effects of WOX5 on H3K9ac, H3K4me3 and H3K27me3 histone modifications and chromatin accessibility by

comparing sorted nuclei from wild-type and *wox5-1* QC cells (Fig. 1A–C,F,G; Appendix Fig. S1B,C). About 89% of all dHMs or dACRs are located between upstream 3 kb of the TSS and downstream 0.5 kb of the TTS of a given gene and only less than 11% in the intergenic regions (Appendix Fig. S6; Dataset EV8). To delineate WOX5 effects on the chromatin of its target genes, we focused on the region upstream 3 kb of the TSS.

This revealed that the directly repressed WOX5 DEGs correlated with WOX5-decreased H3K9ac and H3K4me3, increased H3K27me3, and reduced accessible chromatin domains (Fig. 3E). By contrast, the WOX5-activated direct target genes significantly overlapped with WOX5-increased H3K9ac, H3K4me3, and chromatin accessibility (Fig. 3D,E) unlike the genes not affected by WOX5 as control (Fig. EV4A). Surprisingly, a subset of the directly activated WOX5 DEGs also showed significantly increased H3K27me3 (108/812), in about one-third of which WOX5 also increased activating marks (H3K9ac/H3K4me3) (Fig. 3F), a hall-mark of bivalent chromatin structure (Blanco et al, 2020). Examples include the *ATP-BINDING CASSETTE C4* (*ABCC4*) and *CELLU-LOSE SYNTHASE A2* (*CESA2*) genes, which play crucial roles in regulating root development (Uragami et al, 2024) (Persson et al, 2007). By contrast, we did not find increased bivalent marks at the indirectly upregulated WOX5 DEGs (Fig. EV4B,C). These results indicate that the WOX5-regulated transcription is highly associated with WOX5-dependent histone modifications and chromatin accessibility.

## WOX5 activates expected and unexpected processes in the QC

Among the directly activated WOX5 DEGs, we found an overrepresentation of expected GO terms "cell communication," "root development," and "auxin biosynthetic process" (Savina et al, 2020; Sharma et al, 2024) (Fig. 4A). In addition to these expected functions, however, we also found unexpected terms such as "nitrate transport" (FDR = 1.68e-42) (Fig. 4A; Dataset EV9). The DEGs in this GO term are mainly associated with nitrate uptake, such as *NRT1.1* (Wang et al, 2020), or nitrate storage in the vacuoles, such as *CLCa* (Hodin et al, 2023).

Furthermore, among the putative direct WOX5 target genes (812), we observed that CC-up genes were significantly over-represented in the directly activated WOX5 DEGs (197), and QC-up DEGs in the directly repressed WOX5 DEGs (97) (Fig. 4B). This is in agreement with PCA analysis with our nuclear and published protoplast transcriptomes (Clark et al, 2019; Li et al, 2016), where we found that along PC1, which appears to report differentiation (Fig. 4C,D), the *wox5-1* QC transcriptome was located more towards the undifferentiation direction than the wild-type QC (Fig. 4C). Examples for directly activated WOX5 DEGs in this

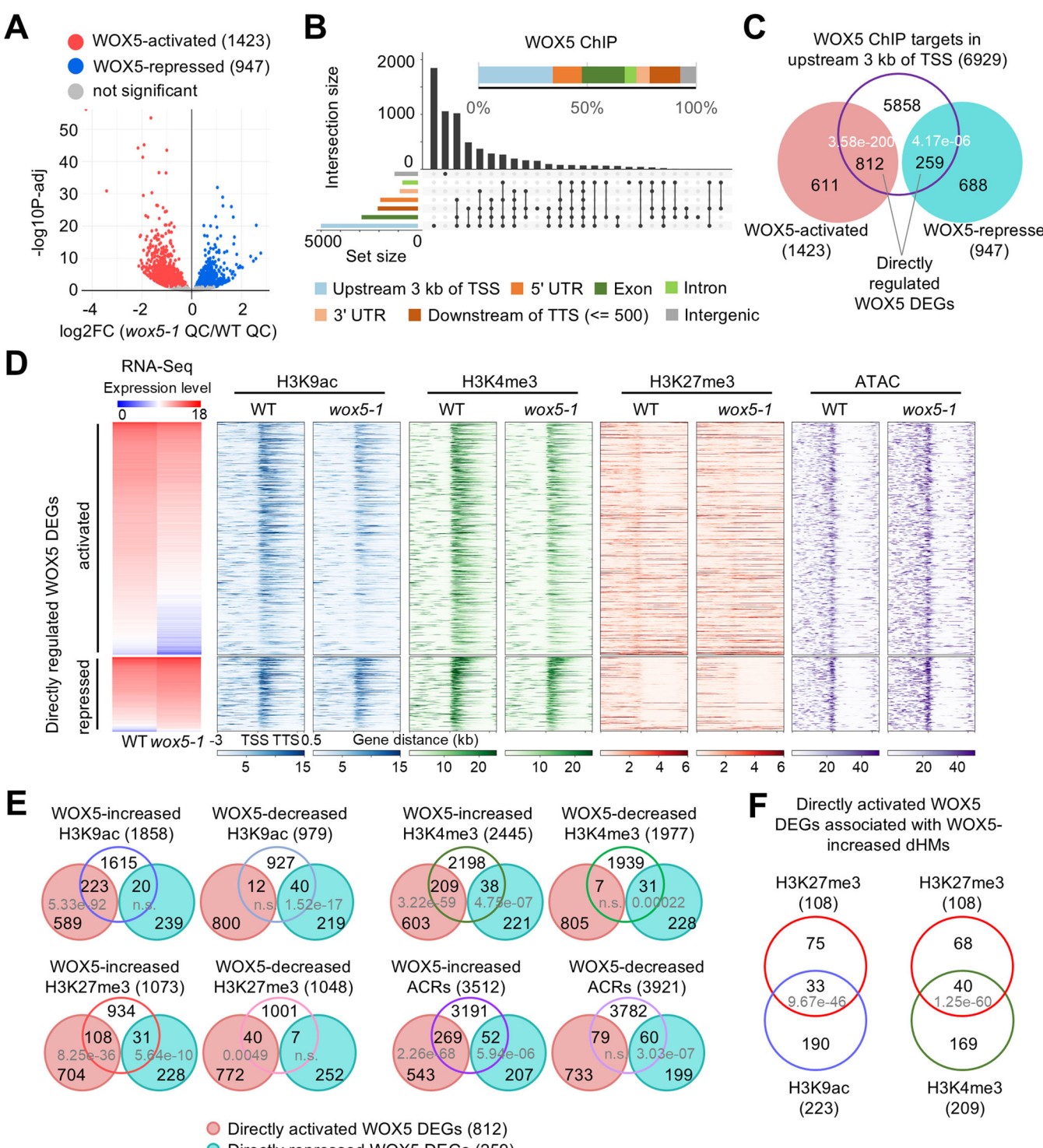

subset are the *AUXIN RESISTANT 3* (*AXR3*) gene promotes trichome differentiation (Perez-Perez et al, 2010), and *NAC046* programmed cell death of the columella root cap, which is the final step of differentiation (Huysmans et al, 2018). The overrepresented GO terms in this subset of "paradoxical" WOX5-activated targets include "epidermal cell differentiation" and "developmental maturation" (Fig. EV5A). To mitigate the limitations of GO term

annotations, we reviewed published studies to address whether genes assigned to a given term had a confirmed corresponding function. Two such examples for the terms "epidermal cell differentiation" and "developmental maturation" are *EXPANSIN A7* (*EXPA7*), which is specifically required for root hair elongation (Lin et al, 2011), and *ATP-BINDING CASSETTE B4* (*ABCB4*), which regulates free IAA levels in the differentiation/maturation

◄ **Figure 3.  Expression levels of directly regulated WOX5 DEGs are associated with chromatin features.**

(A) Volcano plot of the DEGs between *wox5-1* QC and wild-type QC from four biological replicates. Significant DEGs are shown by colored dots. *Padj* <0.05. *Padj*: Benjamini–Hochberg adjusted *P* value. (B) Genomic distribution of WBRs. The bar in the upper part represents the distribution of genomic regions among WOX5-binding fragments. In the Upset plots, the vertical bars represent the numbers of the peaks spanning one or more specific genomic regions, with overlaps indicated by connection lines between the black dots. The horizontal bars indicate the total number of peaks for each genomic annotation, independent of overlaps. (C) Venn diagram of directly regulated WOX5 DEGs with WBRs in upstream 3 kb of TSS. *P* values indicate statistical significance by hypergeometric tests. (D) Chromatin features in the region between 3 kb upstream of TSS and 0.5 kb downstream of the TTS of directly activated and repressed WOX5 DEGs. (E) Venn diagrams of directly regulated WOX5 DEGs and the genes with dHMs and dACRs, respectively. *P* values indicate statistical significance by hypergeometric tests. (F) Venn diagrams of directly regulated WOX5 DEGs with genes associated with WOX5-increased H3K27me3, H3K9ac and H3K4me3, as shown in (E), reveal genes with potential WOX5-induced bivalent chromatin domains. *P* values indicate statistical significance by hypergeometric tests. Source data are available online for this figure.

zone of the root, crucial for maintaining auxin levels, root differentiation, and growth (Kubes et al, 2012).

Importantly, despite the expression levels of this subset of genes being higher in the wild-type QC than in the *wox5-1* mutant, their absolute expression levels were low (Fig. 4E), suggesting that WOX5 only promotes low basal expression levels of these genes. We also observed a similar subset of "paradoxical" indirectly regulated WOX5 DEGs (Fig. EV5B). These results suggest that WOX5 activates a subset of low maturation/differentiation-related transcription levels in the QC.

### Directly activated WOX5 target genes can partially restore QC features

To get first insight into the functional relevance of our data, we selected one representative gene from four interesting GO terms of the directly activated WOX5 DEGs that also display WOX5-increased H3K4me3 levels and chromatin accessibility in the wild-type QC (Table EV6). From the term "nitrate transport," we selected *C-TERMINALLY ENCODED PEPTIDE RECEPTOR 2* (*CEPR2*), a member of the paradoxical DEGs. CEPR2 can phosphorylate nitrate transporter NRT1.2 to mediate nitrate uptake signaling (Zhang et al, 2021) and is also involved in upregulating the nitrate transporter genes *NRT1.1*, *NRT2.1*, and *NRT3.1* in response to nitrogen starvation (Tabata et al, 2014). From "cell wall biogenesis," we selected *CELLULOSE SYNTHASE 1* (*CESA1*), which promotes pectin synthesis and methyl esterification (Zhang et al, 2022). From "root development/cell communication," we selected *SGP1*, mainly expressed in cells with limited or null mitotic activity (Bedhomme et al, 2009). Finally, from "osmotic stress/ response to stress," we selected *EARLY RESPONSE TO DEHY-DRATION 14* (*ERD14*) that responds to osmotic stress, desiccation, and cold (Alsheikh et al, 2003). First, we confirmed higher signals in the wild-type QC with fluorescent reporter genes than in *wox5-1* for all four candidates, confirming their classification as WOX5-upregulated (Fig. 5A–E). We then tested their relevance for WOX5-mediated regulation of the QC by expressing their coding regions from the *WOX5* promoter in the *wox5-1* mutants. Since WOX5 upregulated many genes in the QC and the contribution of each single gene plausibly might be small, we thus used the *QC184:GUS* reporter gene, which is a faithful readout of WOX5 activity in the QC (Fig. 5F,G) (Sarkar et al, 2007). We found that the expression of *CEPR2*, *ERD14*, and *SGP1* partially restored the *QC184:GUS* signal in the QC of *wox5-1*, whereas *pWOX5:CESA1* did not (Fig. 5H–L). These results suggest that *CEPR2*, *ERD14*, and *SGP1* are functionally relevant WOX5 target genes in at least some aspects of QC regulation.

## Discussion

The nature of plant stem cell organizers has remained largely enigmatic. Here, we determine the transcriptional and epigenetic architectures of the root stem cell organizer, the QC, and the function of WOX5 therein. Our results confirm that WOX5 regulates key features of the QC but does so in a broader way than previously known. Furthermore, we reveal unexpected functions of WOX5. Therefore, we propose a novel and broader concept of WOX5 function, including the priming of the QC transcriptome towards mature columella cell fates. In the following, we will discuss the implications of this model for root stem cell niche regulation.

### The nuclear transcriptome as a readout of WOX5 function in the QC

For understanding the transcriptional role of WOX5 in the QC, we have chosen the nuclear transcriptome as the most plausible readout. We note that despite the expected similarities between our nuclear and protoplast-derived transcriptomes, there are also strong differences. Plausibly, the inevitable differences between nuclear and cytoplasmic/cellular transcriptomes due to different mRNA nuclear export or decay rates contribute to these differences (Gaedcke et al, 2022; Palovaara and Weijers, 2019), resulting in a higher nuclear representation of highly dynamic mRNA species, for example, associated with signaling and stress responses (Narsai et al, 2007). In addition, our data suggest that cellular stress and the upregulation of WOX5 triggered by cell wall degradation during protoplast preparation (Birnbaum et al, 2003; Denyer et al, 2019) also contribute. Importantly, we could confirm the majority of DEGs, suggesting that the false positive rate of our approach is low. On the other side, however, we also found evidence that statistical variations between deep sequencing replicates can hamper the recognition of some WOX5-regulated DEGs. Therefore, while the identified DEGs have high confidence levels, we cannot exclude that some WOX5-regulated genes identified by our criteria may have been missed.

### WOX5 affects the chromatin of QC cells

Our previous results suggested that WOX5 represses the transcription of *CDF4* by recruiting an H3K9 deacetylating complex (Pi et al, 2015). Supporting this finding, we detect a significant overlap of WOX5-repressed targets with a decrease of the activating marks H3K9ac and H3K4me3 but with an increase of the repressive mark H3K27me3 in the 3 kb upstream regions.

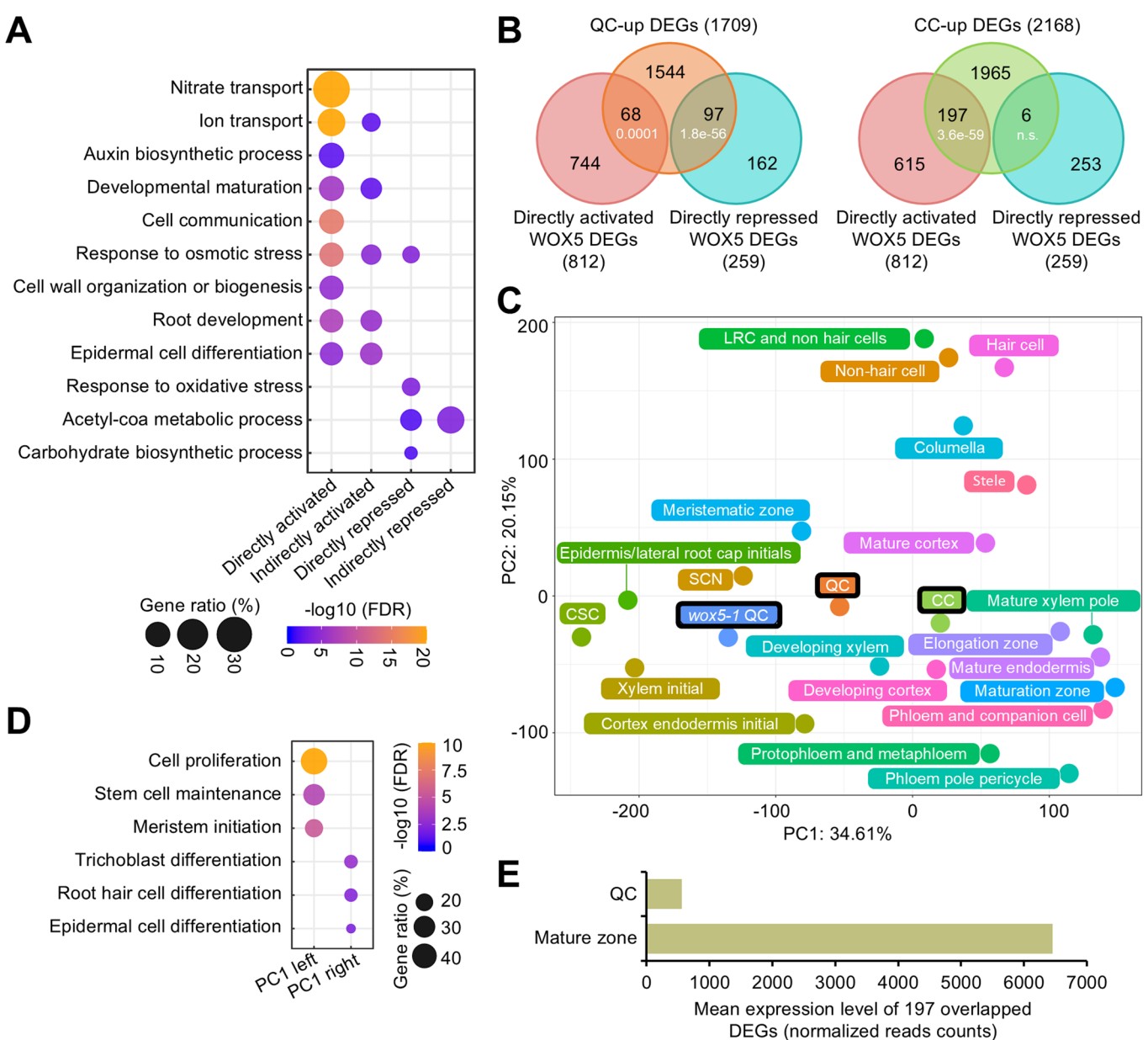

**Figure 4. WOX5-activates genes are associated with cell differentiation/maturation in the QC.**

(A) GO analysis of directly and indirectly regulated WOX5 DEGs. The color gradient indicates –log10 (FDR) by Fisher exact test and Yekutieli correction (FDR < 0.05). (B) Venn diagrams of QC-up/CC-up DEGs and directly regulated WOX5 DEGs. *P* values indicate statistical significance of the overlaps by hypergeometric tests. (C) PCA of transcriptome data from this study (outlined in black) and published root tissues/cells derived from Clark et al (2019) and Li et al (2016). SCN stem cell niche, LRC lateral root cap. (D) GO analysis of the genes in PC1 loading from (C). The color gradient indicates –log10 (FDR) by Fisher exact test and Yekutieli correction (FDR < 0.05). (E) Comparison of the mean expression levels (normalized read counts) of the 197 directly activated "paradoxical" WOX5 DEGs (B) in published QC and root maturation transcriptome data (Li et al, 2016).

However, a function as a transcriptional activator of WOX5 has not previously been reported, and a different mechanism plausibly must be in place. Indeed, among the WOX5-activated direct target genes, we observed a significant increase of H3K9ac and H3K4me3, as expected for transcriptional activation. We note that the increased H3K4me3 levels could simply be an indirect consequence because WOX5 upregulates the Trithorax group H3K4-methyltransferase *SET DOMAIN GROUP 2* (Yao et al,

2013). On the other hand, it is curious that inhibiting its interaction with TPL/HDA19 renders WOX5 largely inactive, whereas a translational fusion with TPL, at least in part, can restore WOX5 activity (Pi et al, 2015). One possible scenario is, therefore, that the WOX5/TPL/HDA19 complex might have an additional, non-canonical function in transcriptional activation. While such a dual function of the WOX5-associated HDA19 has yet to be defined, it is noteworthy that the related histone

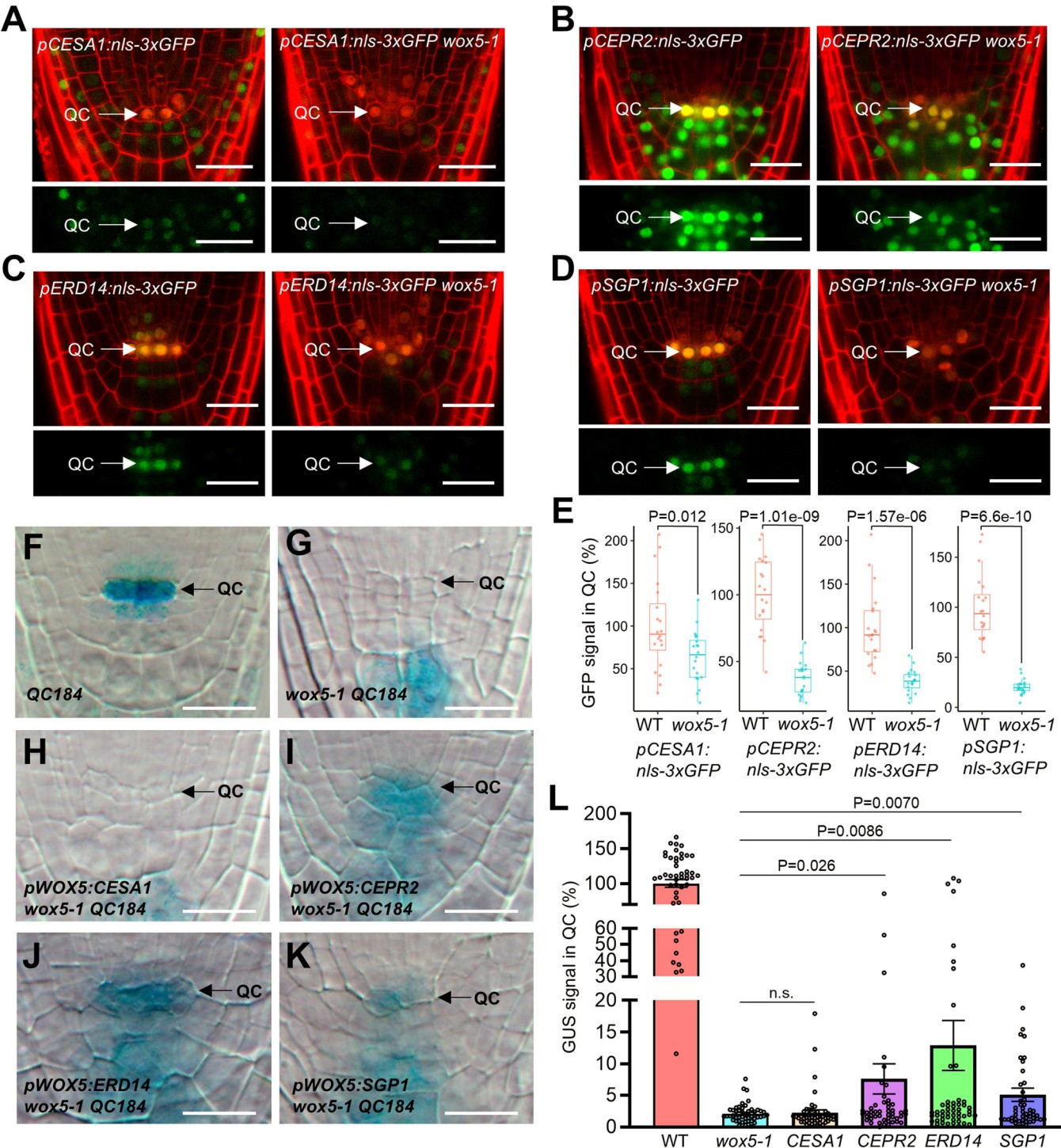

**Figure 5. Expression of *CEPR2*, *ERD14*, and *SGP1* partially restores QC activity in *wox5-1*.**

(A–D) Expression of the indicated reporter lines in *pWOX5:H2B-tdTomato* or *pWOX5:H2B-tdTomato wox5-1* roots. The lower panels show only the GFP channel. *n* = 20, bars = 20 µm. (E) Quantification of GFP signals in the QC of the reporter lines in (A–D). The mean GFP intensity in the wild-type QC was set to 100%. Each box plot shows the median (inside line), 25–75 percentiles (box bottom to top) and whiskers extending to the minimum and maximum values (excluding outliers). *n* = 20 biological replicates/genotype. *P* values by two-sided Student's *t* test are shown. (F–K) *QC184* GUS signal in roots expressing the indicated transgenes from the *pWOX5* promoter in the *wox5-1* background. *QC184* GUS in wild-type is shown as reference (F). Bars = 10 µm. (L) Quantification of GUS signals in the QC of the misexpression lines in (F–K). The mean GUS intensity in the wild-type QC was set to 100%. *n* = 50 biological replicates/genotype. *P* values by two-sided Student's *t* test are shown. Source data are available online for this figure.

deacetylase HDA9 can promote transcription via a process involving the eviction of the H2AZ histone variant (van der Woude et al, 2019). As an alternative hypothesis, WOX5 might reside in alternative protein complexes for transcriptional activation or repression. A similar model has been proposed for the bimodal transcriptional activity of the WOX5 paralog WUSCHEL (WUS), which appears to interact with different sets of transcriptional coregulators to repress or activate transcription in shoot meristem stem cell regulation (Dolzblasz et al, 2016; Ikeda et al, 2009; Leibfried et al, 2005; Yadav et al, 2011; Zhou et al, 2015; Zhou et al, 2018). The different transcriptional modes of WUS appear spatially separated between the organizing center (niche cells) and the shoot meristem stem cells. In contrast to WUS, however, WOX5 would have to interact with different complexes in the same QC cell. Future analysis will address the molecular mechanisms of transcription activation by WOX5.

Notably, we find that WOX5 promotes the establishment of bivalent chromatin domains at a subset of directly activated WOX5 target genes. A phenotypic analysis of root development has been published only for a few examples in this group of genes. The *ABCC4* gene encodes an *Arabidopsis* cytokinin efflux transporter and is crucial in restricting root elongation by controlling the active cytokinin flow (Uragami et al, 2024). The *CESA2* gene encodes a component of a primary cell wall cellulose synthase complex and is essential for maintaining the structural integrity required for root growth (Persson et al, 2007). Bivalent chromatin is thought to maintain genes in a state to quickly respond to activation and repression signals (Bernstein et al, 2006; Macrae et al, 2023; Volker-Albert et al, 2020). Originally, this concept was proposed with activating domains completely embedded in a repressive domain. However, recent studies suggest a broader definition of bivalent chromatin, where activating and repressive domains overlap (Matlik et al, 2023). As a prominent example, the BEND3 transcription factor promotes bivalent chromatin at hundreds of genes in mouse embryonic stem cells (Yakhou et al, 2023). Thus, considering the QC as a stem cell reserve, transcription factors promoting bivalent chromatin in stem cell regulation may be a shared theme between the plant and animal kingdoms.

## WOX5 regulates expected and unexpected pathways

How does WOX5 regulate QC function? Our data reveal that WOX5 has a lesser effect on global transcriptome differences between QC and CC cells but regulates specific key features of the QC, repression of cell divisions, and differentiation and promotion of auxin signaling. In this regard, our data significantly expands previous knowledge of how these features are regulated. For example, previous studies showed that WOX5 promotes QC quiescence by directly repressing the D-type cyclin *CYCD3;3* (Forzani et al, 2014). Here, we show that WOX5 additionally activates the expression of the cyclin-dependent kinase inhibitor genes *KIP-RELATED PROTEIN 1* and *2*, encoding negative regulators of cell division (Verkest et al, 2005), and the repressor of endomitosis *SIAMESE* (*SIM*) (Churchman et al, 2006). Moreover, we found that WOX5 represses the *BOREALIN RELATED INTERACTOR 2* (*BORI2*) gene, which facilitates the segregation of chromosomes during cell division (Komaki et al, 2022). Thus,

WOX5 promotes QC mitotic quiescence through multiple cell cycle-related genes.

Likewise, we found that in addition to the published repression of the differentiation gene *CDF4*, WOX5 also represses *HB-8*, which acts as a differentiation-promoting transcription factor of the vascular meristem (Baima et al, 2001). Moreover, our results showed that WOX5 also activates *BBM* and *RGF8*, which play significant roles in the regulation of root stem cell maintenance (Aida et al, 2004; Fernandez et al, 2015), and *QQS*, which represses starch accumulation (Li et al, 2009). In conclusion, the genomic analysis of WOX5 function in the QC revealed that WOX5 directly regulates multiple components of mitotic quiescence and differentiation inhibition.

On the other hand, we discovered several unexpected WOX5 functions. One unexpected finding is that the GO term "nitrate transport" is one of the most significant networks upregulated by WOX5 in the QC. What could the functional relevance of this regulation be for the QC? Since WOX5 directly upregulates, for example, *CEPR2* and *NTR1.1* expression that promote nitrate uptake (Tabata et al, 2014), one possible scenario is that WOX5 regulates cellular nitrate levels in the QC. Notably, Arabidopsis seedlings grown under low external $NO_3^-$ levels exhibited a *wox5-1* mutant-like phenotype, suggesting that low external $NO_3^-$ and compromised WOX5-mediated $NO_3^-$ uptake may similarly disturb stem cell organizer function. Moreover, external $NO_3^-$ has been found to affect WOX5 function (Wang et al, 2019), suggesting a positive feedback regulation between WOX5 and nitrate. On the other hand, NRT1.1 also facilitates the uptake of auxin (Krouk et al, 2010), and several studies provide evidence that the uptake of shoot-derived auxin contributes to the auxin response maximum in the QC, which is crucial in maintaining the stem cell niche (Benjamins and Scheres, 2008; Della Rovere et al, 2013; Savina et al, 2020). In agreement with the reduced auxin response maximum in the *wox5-1* QC (Sharma et al, 2024; Tian et al, 2014), one alternative hypothesis is, therefore, that WOX5 promotes auxin import into the QC via NRT1.1, in addition to the previously described upregulation of auxin biosynthesis by WOX5 (Savina et al, 2020; Sharma et al, 2024).

Another unexpected finding of our study is that WOX5 upregulates a group of genes in the QC with higher expression levels in mature root cells, such as CCs. This contrasts the view that the QC represents a quiescence stem cell pool (Dolan et al, 1993; van den Berg et al, 1997). How can these two aspects be reconciled? Notably, the expression levels of the WOX5-promoted maturity genes in the QC are much lower than the expression levels in the corresponding mature cell types. Since the QC does not develop, for example, Casparian stripes typical for the endodermis, although WOX5 promotes the expression of genes regulating Casparian stripes, it is plausible that these expression levels are below the threshold of being functionally effective. Therefore, one hypothesis is that WOX5 primes QC cells toward root cell fates that can be rapidly activated when necessary. Two observations are reminiscent of this model. First, a priming mechanism has been proposed for muscle stem cells (satellite cells). These cells exist in a quiescent state (Schultz et al, 1978) but express low levels of the master regulator of muscle differentiation, *Myogenic determination factor 5* (*Myo5*). Only upon injury the *Myo5* network becomes fully active to regenerate new muscle fibers

(Rantanen et al, 1995), and it has been proposed that a low *Myo5* expression is required for this rapid activation (Beauchamp et al, 2000; Ott et al, 1991). Second, WOX5-priming of the QC towards root fates is consistent with the model that the QC serves as a stem cell reserve to replenish root tissue in the case of injury (Matosevich and Efroni, 2021) or to increase root growth with progressive aging (Timilsina et al, 2019; Wein et al, 2020). The low expression level of maturity genes typical for root cell types might thus document a certain commitment of QC cells to produce root tissue. Future studies will be necessary to test this model.

# Methods

### Reagents and tools table

| Reagent/resource | Reference or source | Identifier or catalog number |
|---|---|---|
| **Antibodies** | | |
| Ab-H3K4me3 | Merck | 17-614 |
| Ab-H3K9ac | Merck | 07-352 |
| Ab-H3K27me3 | Merck | 07-449 |
| IgG | Merck | CS200581 |
| IgG | Cell Signaling Technology | #2729 |
| **Oligonucleotides and other sequence-based reagents** | | |
| NEBNext® Multiplex Oligos for Illumina® (Index Primers Set 1) | NEB | E7335S |
| NEBNext® Multiplex Oligos for Illumina® (Index Primers Set 2) | NEB | E7500S |
| NEBNext® Multiplex Oligos for Illumina® (Index Primers Set 3) | NEB | E7710S |
| NEBNext® Multiplex Oligos for Illumina® (Index Primers Set 4) | NEB | E7730S |
| **Chemicals, enzymes, and other reagents** | | |
| Agencourt AMPure XP magnetic beads | Beckman Coulter | A63881 |
| Agilent High Sensitivity DNA Kit | Agilent | 5067-4626 |
| Agilent RNA 6000 Pico Kit | Agilent | 5067-1513 |
| Bio-Mag Plus Concanavalin A coated beads | Polysciences Europe | 86057-3 |
| Buffer Kit, RNase-free | Thermo Fisher | AM9010 |
| EDTA-free Protease inhibitor cocktail | Roche | 05056489001 |
| Illumina Tagment DNA Enzyme and Buffer Small Kit | Illumina | 20034197 |
| pA/G MNase | Addgene | #123461 |
| PureLink™ RNA Micro Scale Kit | Thermo Fisher | 12183016 |

| Reagent/resource | Reference or source | Identifier or catalog number |
|---|---|---|
| NEBNext Single Cell/Low-Input RNA Library Prep Kit for Illumina | NEB | E6420S |
| NEBNext Ultra II DNA Library Prep Kit for Illumina | NEB | E7645S |
| RiboLock RNase Inhibitor | Life Technologies | EO0384 |
| RNaseZap™ RNase Decontamination Solution | Thermo fisher | AM9782 |
| SPRIselect Reagent | Beckman | B23317 |
| **Software** | | |
| Bowtie2 | Langmead and Salzberg, 2012 | https://bowtie-bio.sourceforge.net/bowtie2/index.shtml |
| ChIPPeakAnno | Zhu et al, 2010 | https://bioconductor.org/packages/release/bioc/html/ChIPpeakAnno.html |
| Cutadapt | Martin, 2011 | https://cutadapt.readthedocs.io/en/stable/ |
| DESeq2 | Love et al, 2014 | https://bioconductor.org/packages/release/bioc/html/DESeq2.html |
| Galaxy | Afgan et al, 2018 | https://usegalaxy.eu |
| ImageJ | Schneider et al, 2012 | https://imagej.net/ij/ |
| Limma | Ritchie et al, 2015 | https://bioconductor.org/packages/release/bioc/html/limma.html |
| MACS2 | Feng et al, 2012 | https://github.com/macs3-project/MACS/wiki/Install-macs2 |
| pcaExplorer | Marini and Binder, 2019 | http://shiny.imbei.uni-mainz.de:3838/pcaExplorer/ |
| PlantGSEA | Yi et al, 2013 | http://bioinformatics.cau.edu.cn/PlantGSEA/ |
| R | R-Project | https://www.r-project.org/ |
| Ringo | Toedling et al, 2007 | https://www.bioconductor.org/packages//2.7/bioc/html/Ringo.html |
| STAR | Dobin et al, 2013 | https://github.com/alexdobin/STAR |
| **Other** | | |
| Illumina NovaSeq 6000 | Illumina | |
| S3e Cell Sorter | BIORAD | |

## Plant material and growth conditions

*Arabidopsis thaliana* wild-type and *wox5-1* (SALK038262) plants in the Columbia (Col-0) accession were used. *PET111* was kindly provided by Philip Benfey (Duke University). Seeds were surface sterilized, placed at 4 °C for 3 days, and grown on 0.5× MS media, 1% agar without sucrose. Plates were placed vertically in a growth chamber with 16 h of light and 8 h of darkness at 23 °C.

## Transgenic work

All plasmids were propagated using *E. coli* strain DH5α and verified by sequencing. For sorting QC and CC nuclei, the *pWOX5:H2B-tdTomato* and *PET111::H2B-tdTomato* constructs were cloned using the GreenGate system (Lampropoulos et al, 2013). *pWOX5:H2B-tdTomato* constructs were transformed using the floral dip method (Clough and Bent, 1998) into wild-type plants and introgressed into *wox5-1*. *pOp:H2B-tdTomato* constructs were transformed into *PET111::GFP* to express the fluorophore in the CCs nuclei.

ClonExpress MultiS One Step Cloning Kit (Cat. No. C113-02, Absource Diagnostics GmbH) was used to clone *pCESA1, pCEPR2, pERD14,* and *pSGP1* driving *nls-3xGFP* for constructing reporter lines. Promoter lengths were designed to be around 3000 bp unless a different gene was found upstream to avoid the expression of neighboring genes. The constructs were transformed into *pWOX5:H2B-tdTomato wox5-1/+*, and homozygous transgenic, wild-type, and *wox5-1* genotypes were obtained for confocal imaging at the T3 generation.

Misexpression lines were cloned using the ClonExpress MultiS One Step Cloning Kit (Cat. No. C113-02, Absource Diagnostics GmbH). *CESA1, CEPR2, ERD14,* and *SGP1* CDS were cloned under the *pWOX5* promoter to drive expression in the QC cells. To test GUS expression recovery, the constructs were transformed into *wox5-1 QC184* (Sarkar et al, 2007). Primers for constructs are shown in Table EV7.

## Nuclei isolation and FANS

In all, 5-mm-long root tips of 5-day-old seedlings were collected and ground in liquid nitrogen. FANS was performed as published (Slane and Bayer, 2017) with minor modifications. To release nuclei, samples were incubated for 10 min in nuclei extraction buffer NEB (1% Triton, 0.25 M Sucrose, 10 mM Tris-HCl (pH 8), 10 mM MgCl2, 1× Protease inhibitor cocktail (Sigma-Aldrich), and 1 mM PMSF) at 4 °C. For RNA-Seq experiments, 0.4 U/μl RiboLock RNase Inhibitor (Thermo Fisher) was added to the NEB. The resulting homogenate was filtered through a 40-μm strainer and centrifuged at $2000 \times g$ for 10 min at 4 °C. For FANS, nuclei pellets were resuspended in $1 \times$ PBS with 0.5% BSA (0.4 U/μl RiboLock RNase Inhibitor added for RNA-seq) and sorted under Purity mode using the FL1 (525/30 nm-GFP) and FL2 (586/25 nm-tdTomato) filters of the S3e Cell Sorter (BioRad). Positive nuclei visualized in the R2 sorting gate for the tdTomato signal were collected for further experiments.

## GUS assays

Transgenic plants expressing *CEPR2, CESA1, ERD14,* and *SGP1* CDS under *WOX5* promoter in *wox5-1 QC184* plants were used for GUS staining assays as published (Jefferson et al, 1987) with minor modifications. Root tips of 5 days seedlings were cut and incubated in staining buffer (500 mM NaPO$_4$ pH 7.0, 10% Triton X-100, 100 mM K$_3$Fe(CN)$_6$, 100 mM K$_4$[Fe(CN)$_6$], 50 mM x-GlcA) at 37 °C for 24 h in dark. Roots were transferred to 70% ethanol and incubated at 37 °C for 1 h. Finally, samples were mounted in chloral hydrate clearing solution containing glycerol and imaged using a Zeiss Axioskop 2 microscope. The staining intensity was measured with ImageJ (Schneider et al, 2012).

## ChIP-Chip assay

The ChIP for identifying WOX5-binding sites was performed as described before (Forzani et al, 2014; Pi et al, 2015) with minor modifications. Two biological replicates were processed for each genotype of *35S:WOX5-GR* and *wox5-1*. Precipitated DNA was eluted with $3 \times 100\,\mu L$ elution buffer (0.1 M glycine, 0.5 M NaCl, 0.05% Tween-20, pH 2.8) and neutralized by adding 50 μL 1 M Tris (pH 9.0) for each elution step (Kaufmann et al, 2010). Proteinase K (Roche) was added to a final concentration of 0.5 mg/mL to elute DNA, which was then incubated at 65 °C overnight. After that, equal amounts of proteinase K were added and incubated at 65 °C for 4 h. DNA was precipitated by adding 2.5× volume of 100% ethanol, 10% sodium acetate (3 M, pH 5.4), and 1 μL glycogen at −20 °C overnight and collected by centrifugation at $13,000 \times g$ for 30 min at 4 °C. The pellet was dissolved in 12 to 15 μL H$_2$O. 10 μL of ChIP-DNA and 10 ng input DNA were amplified using the WGA2 amplification kit (Sigma-Aldrich). Amplified DNA was purified with the PCR purification kit (Qiagen) and was eluted in 50 μL H$_2$O. DNA was labeled and hybridized using SureHyb hybridization chambers according to the Agilent labeling and hybridization protocol (Agilent Mammalian ChIP-on-chip protocol, Version 10.2, 2011).

## ChIP-Chip bioinformatic analysis

ChIP-Chip data was processed using Ringo software (Toedling et al, 2007) and data were analyzed on R version 2.15.1 (http://www.r-project.org), using packages from the Bioconductor project (Gentleman et al, 2004). This study used a high stringency parameter "thresholds = 2, distCutOff = 250, minProbesInRow = 5" to identify ChIP-enriched regions. *CDF4* can be found with stringency setting "thresholds = 2, distCutOff = 250, minProbesInRow = 4". All these parameters are more stringent than the standard ones ("distCutOff = 600, minProbesInRow = 3") (Toedling et al, 2007). ChIP-Chip peaks were annotated using the annotatePeakInBatch (multiple = TRUE, select = "all") function in the ChIPPeakAnno package (Zhu et al, 2010). This function allowed for annotating overlapping genomic features for each ChIP peak within the specified genomic region.

## Preparation of RNA-seq libraries, sequencing, and RT-qPCR validation

Five thousand positive nuclei were sorted for each of the four biological replicates and directly sorted in lysis buffer supplemented with β-mercaptoethanol. RNA was extracted with the PureLink® kit (Thermo Fisher). According to the manufacturer's instructions, RNA-seq libraries were prepared with the NEBNext® Single Cell/Low Input RNA Library Prep Kit for Illumina® kit (NEB). Overall, 15 PCR cycles were used for cDNA amplification, and 10 were used for library amplification. We selected fragment sizes in the 150–500 bp range with AMPure XP beads (Beckman), and a sharper peak of around 300 bp was visible in an Agilent High Sensitivity DNA assay for all sample libraries. The libraries were then sequenced on an Illumina NovaSeq using 150 bp pair-end reads. For RT-qPCR validation, 5000 positive nuclei were collected from two separate biological replicates and prepared for libraries with the same protocol for RNA-Seq library construction. Primers for RT-qPCR are shown in Table EV7.

## RNA-seq bioinformatic analysis

For RNA-Seq, comparisons were made between wild-type CCs and wild-type QC, and between *wox5-1* and wild-type QC. FASTQ files were processed and analyzed on the European Galaxy server (https://usegalaxy.eu) (Afgan et al, 2018). The quality of raw reads was assessed by fastQC. Paired-end reads were trimmed by Cutadapt (Martin, 2011), cropping "-30 bases" from the end of reads, using "a quality cutoff -q20, minimum length of 20 bp", and removing Illumina adapters (AGATCGGAAGAG). Trimmed reads were aligned to the TAIR10 genome using RNA STAR (v2.7.6a) (Dobin et al, 2013), and reads were counted by FeatureCounts (Liao et al, 2014) using default parameters. Differentially expressed genes were identified by DESeq2 (v2.11.40.6) (Love et al, 2014).

## Preparation of CUT&RUN libraries and sequencing

Five thousand sorted nuclei for each biological replicate of H3K9ac, H3K4me3, H3K27me3, and IgG were collected in 1x PBS with 0.5% BSA, pelleted at $2000 \times g$ for 10 min at 4 °C, and the supernatant was removed. Standard CUT&RUN was performed as described (Skene and Henikoff, 2017) with some modifications. In summary, 10 μl of Bio-Mag Plus Concanavalin A were added to nuclei and incubated for 10 min at room temperature. Blocking was performed with 1.5 mL of blocking buffer for 5 min at room temperature. Ab-H3K9ac (Cat. No. 07-352, Merck), ab-H3K4me3 (Cat. No. 17-614, Merck), ab-H3K27me3 (Cat. No. 07-449, Merck) and IgG (Cat. No. CS200581, Merck, and #2729, Cell Signaling) were added in a concentration of 1:100 and incubated for 2 h at 4 °C. For DNA-protein complex digestion, pA/G MNase purified in our laboratory from plasmid #123461 (Addgene) was added, and samples were incubated for 1 h at 4 °C. Activation of MNase was achieved by incorporating 2 μl 100 mM $CaCl_2$ for 30 min at 0 °C. The reaction finished with the addition of 100 μl 2× STOP buffer. Samples were incubated for 10 min at 37 °C, and total DNA was released by the disruption of the nuclei with 2 μl 10% SDS and 2.5 μl proteinase K (20 mg/ml). DNA was extracted by Phenol/Chloroform/Isoamyl alcohol (25:24:1) solution as described (Zheng and Gehring, 2019). Following the manufacturer's instructions, the library was performed with the NEBNext Ultra II DNA Library Prep Kit for Illumina. In total, 14–15 PCR cycles were used for DNA amplification, and fragments were size selected between 150 and 1000 bp with AMPure XP beads (Beckman). DNA fragment sizes were checked with an Agilent High Sensitivity DNA assay kit, and samples were then sequenced on an Illumina NovaSeq using 150 bp pair-end reads.

## CUT&RUN bioinformatic analysis

For CC and QC CUT&RUN, wild-type CCs and wild-type QC were compared. For *wox5-1* and wild-type QC CUT&RUN, *wox5-1* and wild-type QC were compared. FASTQ files were processed and analyzed on the European Galaxy server (https://usegalaxy.eu) (Afgan et al, 2018). The quality of raw reads was assessed by fastQC. Paired-end reads were trimmed by Cutadapt (Martin, 2011), using a quality "cutoff -q20, minimum length of 20 bp", and removing Illumina adapters (AGATCGGAAGAG). Trimmed reads were aligned to the *E.coli* genome for the spike-in normalization and to the TAIR10 genome using Bowtie (v2.4.2) (Langmead and

Salzberg, 2012) with modifications to allow mapped paired-end reads "up to 700 bp, 10 bp as a minimum, and very-sensitive" parameters. Mitochondrial and chloroplast reads were removed using BamTools Filter with the parameters "isProperPair true, mapQuality ≥20, and -reference! Mt and ! Pt". PCR duplicates were removed using PicardTools MarkDuplicates. Narrow (H3K9ac and H3K4me3) and broad (H3K27me3) peak calling were performed for individual replicates and pooled replicates using MACS2 (Feng et al, 2012) with a "–nomodel" parameter and IgG files as a control. Then we concatenated all peaks from MACS2 BED files in wild-type and *wox5-1* or CC and QC samples, sorted by chromosome and ascendant start position, and combined the nearby intervals into a single one to use as a GTF input gene annotation file for FeatureCounts (Liao et al, 2014). Differential CUT&RUN peaks were identified by Limma (v3.50.0) (Ritchie et al, 2015) by a cutoff *P*adj <0.05.

## Preparation of ATAC-Seq libraries and sequencing

Ten thousand positive nuclei were collected from each of the three biological replicates. Nuclei were sorted in 1× PBS with 0.5% BSA, pelleted at $2000 \times g$ for 10 min at 4 °C, and the supernatant was removed. ATAC-seq was performed using the Illumina Tagment DNA Enzyme and Buffer Small Kit (Illumina). Segmentation was accomplished at 37 °C for 30 min. DNA fragments were purified with Monarch PCR & DNA cleanup kit (NEB). Tagmented DNA was amplified using NEBNext High-Fidelity 2× PCR Master Mix for 10 to 11 cycles, and quality was verified with an Agilent High Sensitivity DNA assay kit. The libraries were sequenced on an Illumina NovaSeq using 150 bp pair-end reads.

## ATAC-seq bioinformatic analysis

For ATAC-Seq, comparisons were made between wild-type CCs and wild-type QC, and between *wox5-1* and wild-type QC. FASTQ files were processed and analyzed on the European Galaxy server (https://usegalaxy.eu) (Afgan et al, 2018). The quality of raw reads was assessed by fastQC. Paired-end reads were trimmed by Cutadapt (Martin, 2011), cropping "-30" bases from the end of reads, using a quality "cutoff -q20, minimum length of 20 bp" and removing Illumina adapters (CTGTCTCTTATA). Trimmed reads were aligned to the TAIR10 genome using Bowtie (v2.4.2) (Langmead and Salzberg, 2012), with modifications to allow mapped paired-end reads "up to 1000 bp, -dovetail and -very-sensitive" parameters. Mitochondrial and chloroplast reads were removed using BamTools Filter with the parameters "isProperPair true, mapQuality ≥ 20, and -reference !Mt and !Pt". PCR duplicates were then removed using PicardTools MarkDuplicates. Uniquely mapped reads were "(+ 4 bp/-5 bp) shifted" by ATAC-shift using the alignment Sieve tool from DeepTools2 (Ramírez et al, 2016) and filtered to a "maximum fragment length of 110 bp". Filtered reads were converted from BAM to BED for peak calling. Narrow peak calling was performed for individual replicates and pooled replicates using MACS2 (Feng et al, 2012) with "a –nomodel, –extsize 200 and -shift 100" to get the peaks around the nucleosomes. Then we concatenated all peaks from MACS2 BED files in wild-type and *wox5-1* or CC and QC samples, sorted by chromosome and ascendant start position, and combined the nearby intervals into a single one to use as a GTF input gene

annotation file for FeatureCounts (Liao et al, 2014). Differential peaks were identified by Limma (v3.50.0) (Ritchie et al, 2015) by a cutoff *P*adj <0.05.

## Peaks annotation, PCA, and GO analysis

ATAC-Seq and CUT&RUN peaks were annotated using the annotatePeakInBatch (multiple=TRUE, select = "all") function in the ChIPPeakAnno package (Zhu et al, 2010). This function allowed for annotating overlapping genomic features for each peak within the specified genomic region. PCAs were generated with pcaExplorer (Marini and Binder, 2019). The published FACS root tissues/cells (Clark et al, 2019; Li et al, 2016) were first normalized to corresponding QC samples and then used for PCA analysis together with FANS data in this study.

The GO enrichment analyses were performed with the PlantGSEA database (Yi et al, 2013), and redundant terms were removed using Revigo with default parameters (Supek et al, 2011).

## Confocal microscopy

Root tips of 5 days seedlings were dissected and mounted in 10 ng/mL propidium iodide (PI) and imaged using a ZEISS LSM700 microscope with 488 nm (GFP) and 555 nm (PI and tdTomato) lasers. We observed that tdTomato (ex 554 nm/em 581 nm) signal localizes the QC bleed-through into the GFP channel. We applied a series of linear algebraic operations to subtract the amount of tdTomato bleed-through in each picture to obtain the corrected GFP signal.

## Data availability

The ChIP-Chip data generated in this study have been deposited in the GEO database under accession code GSE270459. The RNA-Seq, ATAC-Seq, and histone CUT&RUN data generated in this study have been deposited in the BioProject under BioProject ID PRJNA1127817 and PRJNA1132792.

The source data of this paper are collected in the following database record: biostudies:S-SCDT-10_1038-S44318-024-00302-2.

## Peer review information

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

## Acknowledgements

The authors are grateful to Philip Benfey (Duke University) for *PET111* seeds, Aileen Riesle and Daria Onichtchouk (Freiburg University) for help with ATAC-Seq, Leily Rabbani and Thomas Manke (Max Planck Institute of Immunobiology and Epigenetics) with CUT&RUN, and Efthimia Yousefi (Freiburg University) with nuclei sorting. The authors acknowledge the support of the Freiburg Galaxy Team: Björn Grüning, Bioinformatics, University of Freiburg (Germany), funded by the German Federal Ministry of Education and Research BMBF grant 031 A538A de.NBI-RBC and the Ministry of Science, Research and the Arts Baden-Württemberg (MWK) within the framework of LIBIS/de.NBI Freiburg. This work was funded by grants from the German Research Foundation (DFG) under Germany's Excellence Strategy (CIBSS—EXC-2189) and La606/18 (Project number 329370126) to TL, Shandong Agricultural University Talent Introduction Start-up Funding and the Germany's Excellence Strategy CIBSS Launchpad Programme funding (CIBSS—EXC-2189) to NZ, and the Research Training Group GRK2344 "MeInBio" (to NZ, PB, and TL).

## Author contributions

**Ning Zhang**: Conceptualization; Resources; Data curation; Software; Formal analysis; Supervision; Funding acquisition; Validation; Investigation; Visualization; Methodology; Writing—original draft; Project administration; Writing—review and editing. **Pamela Bitterli**: Conceptualization; Resources; Data curation; Software; Formal analysis; Validation; Investigation; Visualization; Methodology; Writing—original draft; Project administration; Writing—review and editing. **Peter Oluoch**: Investigation; Methodology. **Marita Hermann**: Investigation; Methodology. **Ernst Aichinger**: Investigation; Methodology. **Edwin P Groot**: Data curation; Software; Formal analysis; Investigation; Methodology. **Thomas Laux**: Conceptualization; Resources; Data curation; Formal analysis; Supervision; Funding acquisition; Methodology; Writing—original draft; Project administration; Writing—review and editing.

Source data underlying figure panels in this paper may have individual authorship assigned. Where available, figure panel/source data authorship is listed in the following database record: biostudies:S-SCDT-10_1038-S44318-024-00302-2.

## Funding

## Disclosure and competing interests statement

The authors declare no competing interests.

# Expanded View Figures

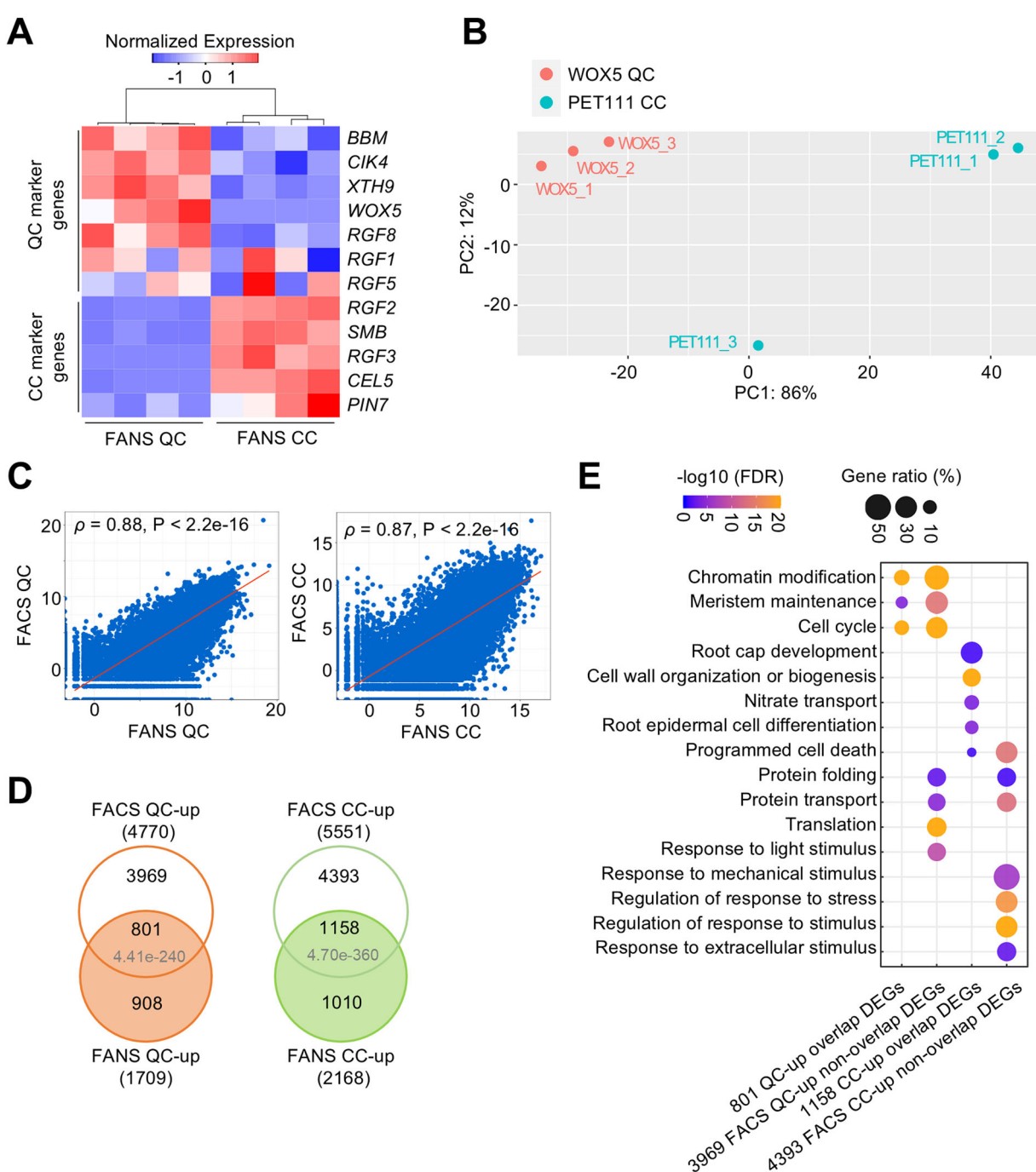

**Figure EV1.  Comparing FACS and FANS data of CC and QC.**

(A) Heatmap of the expression values of published QC- and CC-associated genes in the QC-up and CC-up FANS transcriptome data. Four biological replicates are compared. The hierarchical clustering of the samples is based on Euclidean distance. (B) PCA of published FACS transcriptomes of QC and CCs (Li et al, 2016). Each dot represents one biological replicate. The first two components with the largest contributions are shown. Replicate PET111_3 was considered an outlier and removed for further analysis. (C) Pairwise correlation of QC and CC transcriptomes from this study (FANS) and from published sorted protoplast data (FACS, Li et al, 2016). The red line represents a linear regression. Spearman's rank correlation coefficients ($\rho$) and P values are shown. Correlation P value between FACS QC and FANS QC is < 2.2e-16. Correlation P value between FACS CC and FANS CC is < 2.2e-16. (D) Venn diagrams of FANS DEGs (CC/QC) from this study and FACS DEGs (CC/QC) from (Li et al, 2016). The P values indicate the statistical significances of the overlaps by hypergeometric tests. (E) GO analysis of shared and non-shared DEGs from FANS (this study) and FACS (Li et al, 2016) transcriptomes as shown in (C). The color gradient indicates −log10 (FDR) by Fisher exact test and Yekutieli correction (FDR < 0.05).

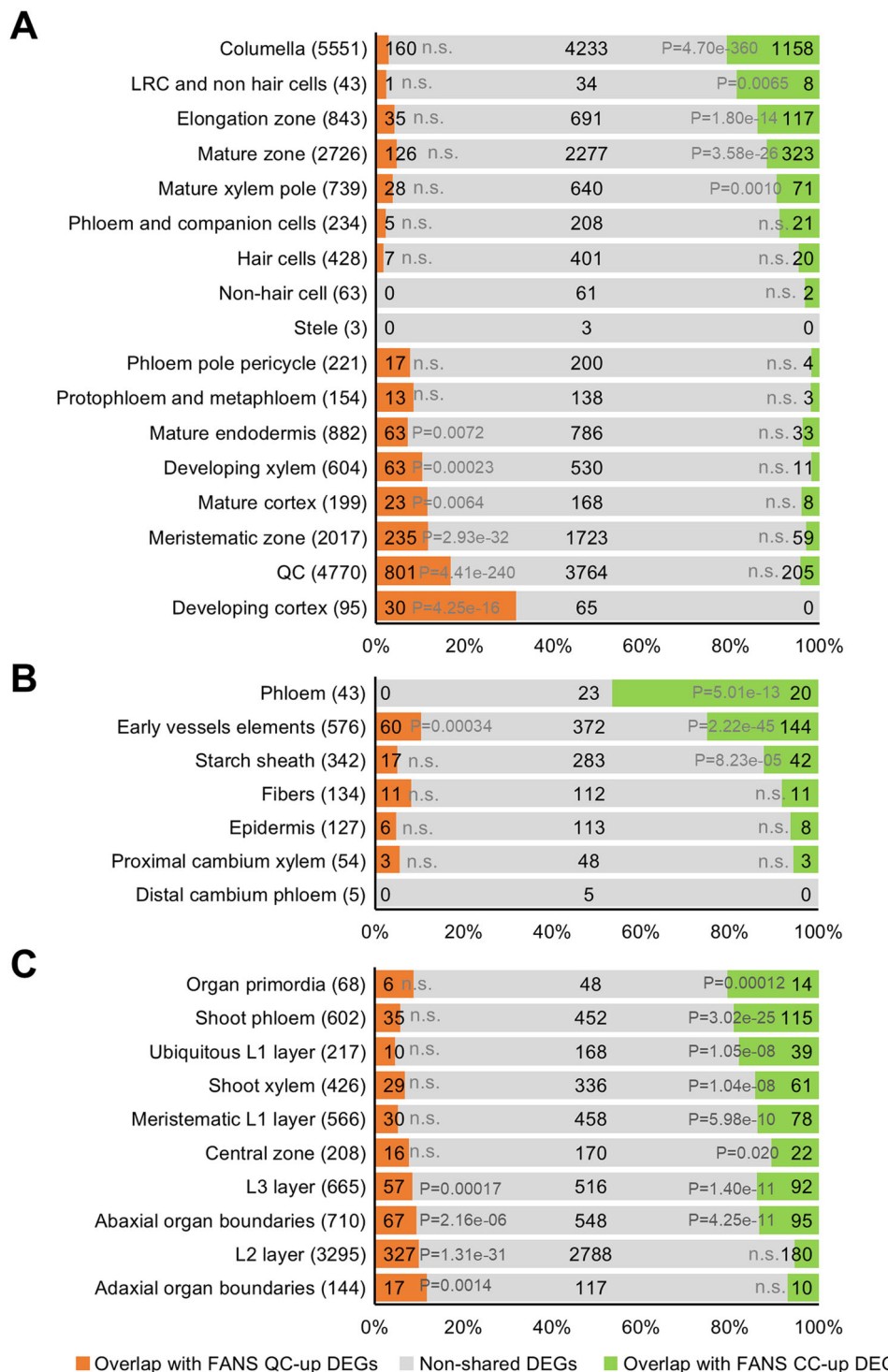

**Figure EV2. Comparison of FANS CC and QC DEGs from this study with published data of different tissues.**

(A–C) Comparison of FANS-derived DEGs from this study with tissue-specific DEGs from FACS of root cells (Li et al, 2016) (A), FANS of vascular cells (Shi et al, 2021) (B) and FACS of shoot apical meristem cells regions (Yadav et al, 2014) (C). The numbers of the published tissue-specific DEGs are shown in brackets. The numbers of shared DEGs and non-shared DEGs are indicated. P values by hypergeometric test are shown; n.s., not significant.

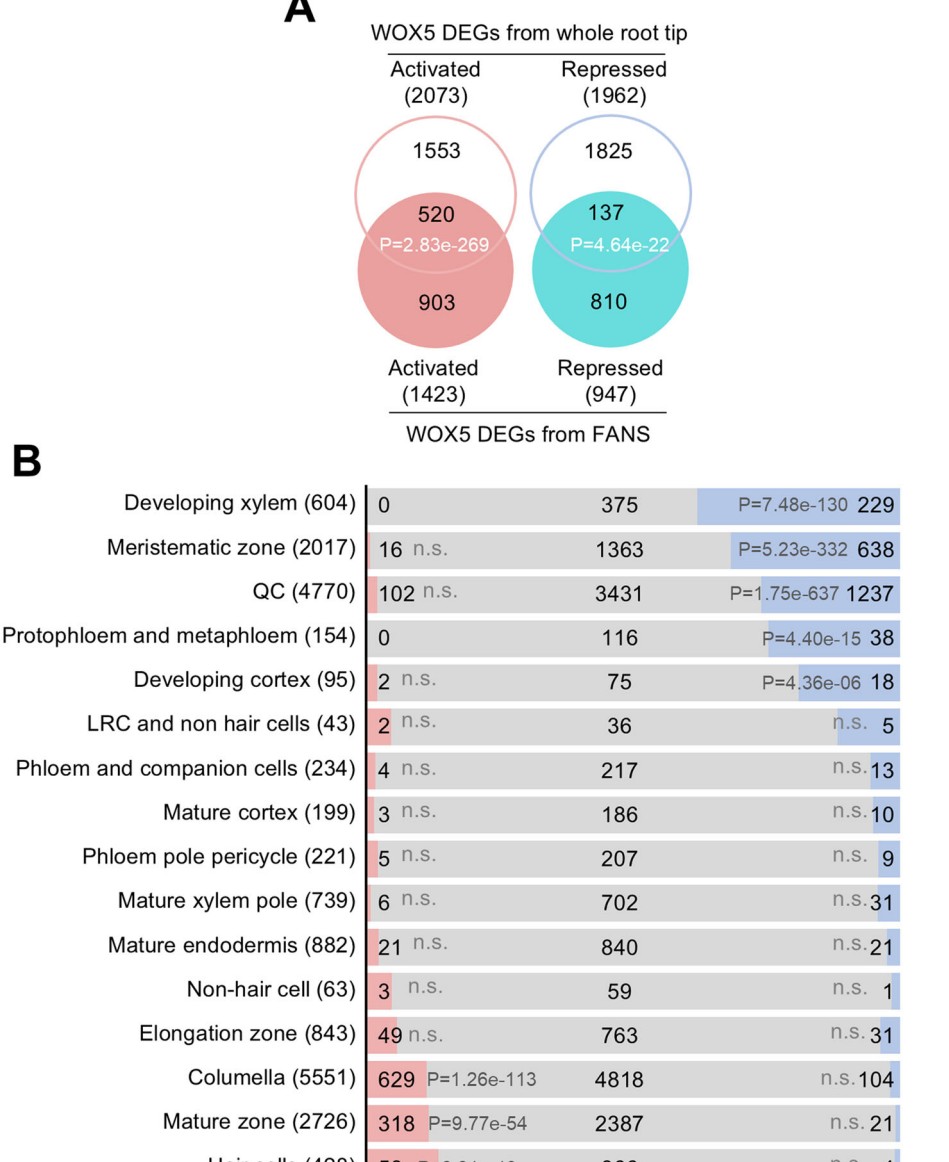

**Figure EV3. Comparison of WOX5 DEGs from sorted nuclei and the whole-root tip.**

(A) Venn diagrams of WOX5 DEGs from FANS transcriptomes (this study) and whole-root-tip transcriptome (Padj <0.05, FC ± 1.5, Clark et al, 2020). P values indicate the statistical significance of the overlaps by hypergeometric tests. (B) Comparison of the WOX5 DEGs derived from whole-root tips (Clark et al, 2020), which are not shared with our FANS data, with FACS-derived DEGs of different root regions from (Li et al, 2016). The numbers of the published tissue-specific DEGs are shown in brackets. The numbers of shared DEGs and non-shared DEGs are indicated. P values by hypergeometric test are shown; n.s., not significant.

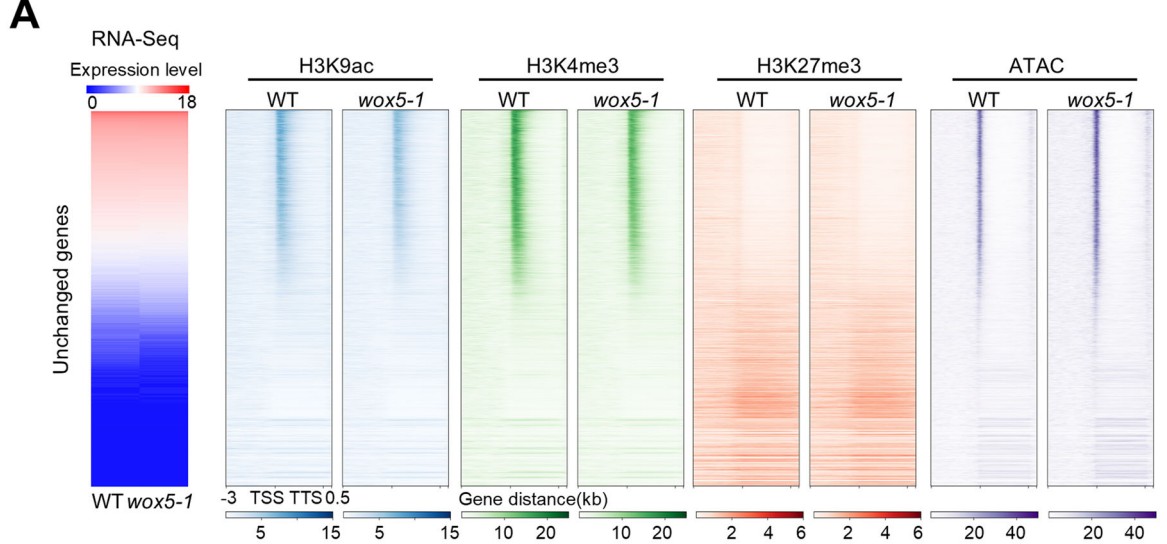

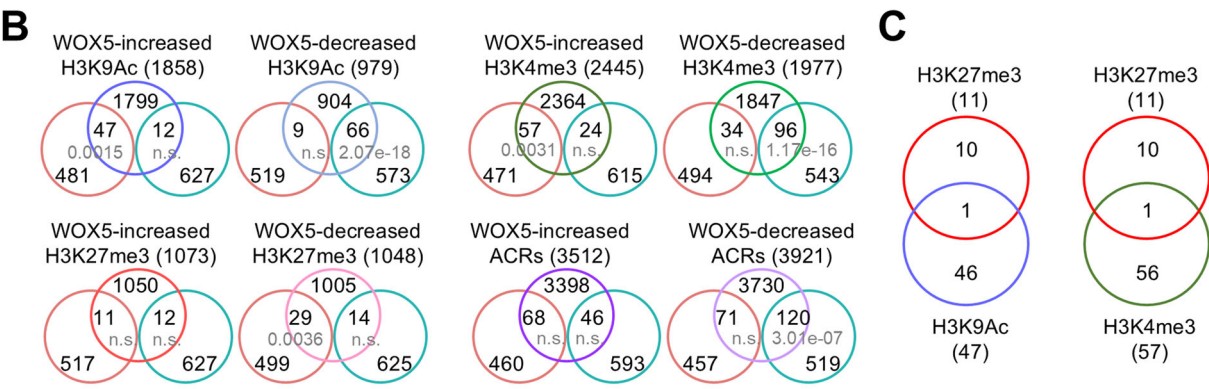

**Figure EV4.** **Integrated analysis of indirect WOX5 DEGs, dHMs, and dACRs between *wox5-1* and WT QC.**

(A) Histone modification and chromatin accessibility profiles between 3 kb upstream of TSS and downstream 0.5 kb of TTS of unchanged genes (unchanged expression between *wox5-1* and wild-type QC). (B) Venn diagrams of indirectly regulated WOX5 DEGs and the genes assigned to dHMs and dACRs. *P* values indicate the statistical significance of the overlaps by hypergeometric tests. (C) Comparison of WOX5-increased activating and repressive dHMs reveals no enrichment of bivalent marks in indirect regulated WOX5 DEGs.

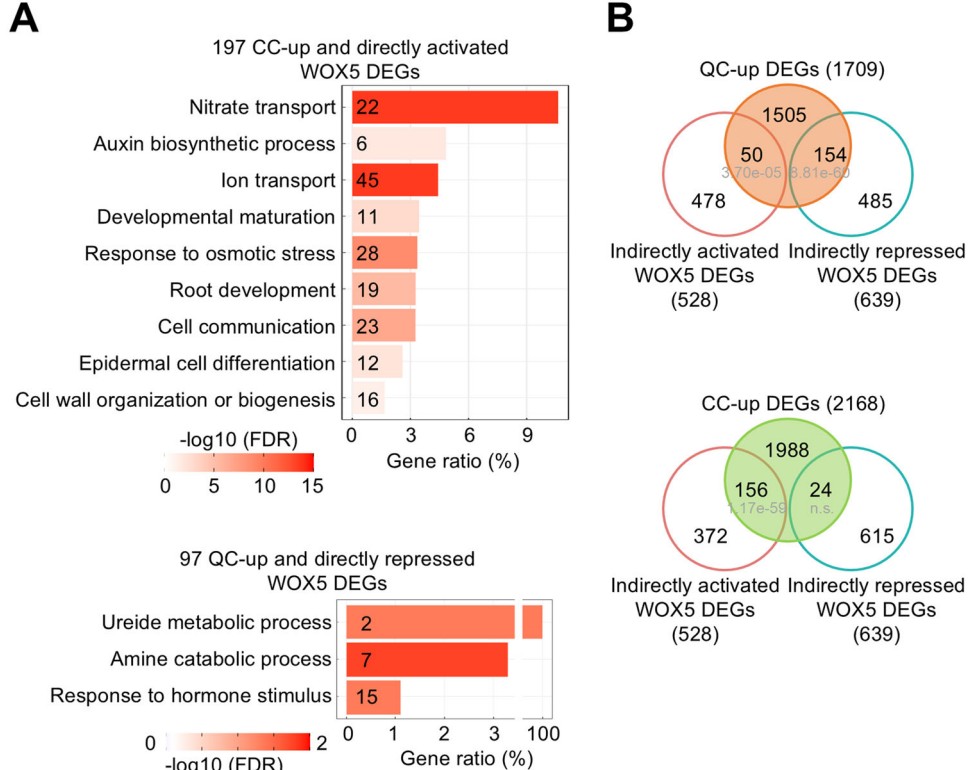

**Figure EV5.  Integrated analysis of QC-up/CC-up DEGs and WOX5 DEGs.**

(A) GO analysis of the "paradoxical" gene subset. The color gradient indicates –log10 (FDR) by Fisher exact test and Yekutieli correction (FDR < 0.05). The number of genes in each term is indicated. (B) Venn diagrams of QC-up/CC-up DEGs and indirectly regulated WOX5 DEGs. *P* values indicate the statistical significance of the overlaps by hypergeometric tests.

