## [Peer Review File · The EMBO Journal]

Deciphering the molecular logic of WOX5 function in the root stem cell organizer

Thomas Laux, Ning Zhang, Pamela Bitterli, Peter Oluoch, Marita Hermann, Ernst Aichinger, and Edwin Groot

Corresponding authors: Thomas Laux (laux@biologie.uni-freiburg.de), Ning Zhang (nzhang@sdau.edu.cn)

Review Timeline:

Submission Date:	9th Jun 24
Editorial Decision:	17th Jul 24
Revision Received:	8th Sep 24
Editorial Decision:	25th Sep 24
Revision Received:	17th Oct 24
Accepted:	25th Oct 24

Editor: Ieva Gailite

Transaction Report:

Dear Dr. Laux,

Thank you for submitting your manuscript for consideration by the EMBO Journal. We have now received comments from a full set of reviewers, which are included below for your information.

As you will see from the reports, reviewers #1-2 are very positive in their assessment and ask for rather minor clarifications and additions to the study. Reviewer #3 also find the study of interest, while raising several aspects that contradict previous literature and would benefit from further exploration. Based on these positive assessments, I invite you to address these concerns in a revised version of the manuscript. I think it would be helpful to discuss the revision in more detail via email or phone/videoconferencing, in particular regarding the feasibility of the requests raised by reviewer #3. I should also add that it is The EMBO Journal policy to allow only a single major round of revision and that it is therefore important to resolve the main concerns at this stage.

We generally allow three months as standard revision time, which can be extended to six months in the case of major revisions. Should you foresee a problem in meeting this deadline, please let us know in advance to discuss an extension. As a matter of policy, competing manuscripts published during this period will not negatively impact on our assessment of the conceptual advance presented by your study. However, please contact me as soon as possible upon publication of any related work to discuss the appropriate course of action.

When preparing your letter of response to the referees' comments, please bear in mind that this will form part of the Review Process File and will therefore be available online to the community. For more details on our Transparent Editorial Process, please visit our website: <https://www.embopress.org/page/journal/14602075/authorguide#transparentprocess>. Please also see the attached instructions for further guidelines on preparation of the revised manuscript.

Please feel free to contact me if you have any further questions regarding the revision. Thank you for the opportunity to consider your work for publication. I look forward to discussing your revision.

With best regards,

leva

leva Gailite, PhD
Senior Scientific Editor
The EMBO Journal
Meyerhofstrasse 1
D-69117 Heidelberg
Tel: +4962218891309
i.gailite@embojournal.org

- a point-by-point response to the referees' comments, with a detailed description of the changes made (as a word file).
- a word file of the manuscript text.
- individual production quality figure files (one file per figure)
- a complete author checklist, which you can download from our author guidelines (<https://www.embopress.org/page/journal/14602075/authorguide>).

- Expanded View files (replacing Supplementary Information)

- a Reagents and Tools Table as part of the Methods section, which can be downloaded from our author guidelines

(<https://www.embopress.org/page/journal/14602075/authorguide#structuredmethods>)

We realize that it is difficult to revise to a specific deadline. In the interest of protecting the conceptual advance provided by the work, we recommend a revision within 3 months (15th Oct 2024). Please discuss the revision progress ahead of this time with the editor if you require more time to complete the revisions.

Referee #1:

The manuscript entitled "Deciphering the molecular logic of WOX5 function in the root stem cell organizer" by Zhang et al. reported the mechanism of WOX5-mediated QC functions in the root tip stem cell niche. The authors used the fluorescent-activated nuclei sorting and revealed the genes/pathways regulated by WOX5 and the accompanied epigenetic landscapes. Based on these data, they found that WOX5 served as both transcription activator and repressor, and identified a new pathway related to nitrate uptake as a downstream of WOX5. Furthermore, the data supported the idea that QC might function as a stem cell reserve. Overall, the manuscript provides many interesting data and will surely improve our understanding of the QC role in the root tip stem cell niche. The manuscript is well organized. I have only two minor suggestions for the authors to improve the manuscript.

Specific points:

- 1) The "Bivalent" chromatin domains discovered in QC are interesting. Some of the data were organized in Figure 3 and Supplemental data. It will be interesting to show some specific cases with bivalent epigenetic modification loci in Figure 3 (or other places in the main text), and have a discussion on some specific genes related to stem cells/QC or other pathways.
- 2) Why did the authors choose H3K4me3 and H3K9ac as the epigenetic markers for gene activation but not choose H3K36me3? In plants, H3K36me3 is typically associated with gene activation. The authors may have an explanation in the main text.

Referee #2:

Major comments:

1. For the GO enrichments, were these performed against the TAIR10 genome or against only the detected transcripts from each experiment? Please clarify how the GO analyses were performed in the methods section (lines 544-545).
2. All sequence data should be deposited at a public repository with accession numbers.
3. Additional discussion on which "cell cycle" genes are bound and regulated by WOX5 would be useful to tie into the introduction, which discusses the concept of mitotically active/inactive cell states. For example, are there particular classes of CYC genes (or other cell cycle regulators) that are bound and regulated by WOX5?

Minor comments:

1. Line 27: replace "basic" with "basal".
2. Line 28-29: suggest re-wording to "These data support a model for QC cells as both reserve stem cells and primed cells for prospective progenitor fates".
3. Line 58-59 needs further elaborated/clarified. A study from Clark et al., 2019 Nature Comm (<https://doi.org/10.1038/s41467-019-13132-2>) characterized stem cell ubiquitous gene expression in Arabidopsis roots and therefore may be a better resource for identifying QC-enriched/specific genes rather than enhancer trap lines that are not mapped.
4. Line 71: suggest re-wording to "WOX5 expression is also positively regulated by auxin, suggesting a feed forward loop between WOX5 and auxin in QC cells".
5. Line 72: suggest changing "dependent and independent" to "coordinated" and "parallel".
6. Line 76: define EAR acronym.
7. Line 146: suggest re-wording to "We also identified enriched GO biological processes in QC cells that have not previously

been linked with root stem cells."

8. Line 173: was this observed "slightly higher" frequency of genes enriched for both chromatin marks more than random chance? A statistical test for this enrichment could help support this result.

9. Line 185: replace "Our" with "The".

10. Lines 204-205: For clarity, perhaps re-word to indicate that these genes are both bound and regulated by WOX5.

11. Lines 247 and 255: GO terms are notorious for being incorrect sometimes or based on limited understanding of particular gene functions.

12. Line 254: change "basic" to "basal".

Referee #3:

The manuscript by Zhang et al. "Deciphering the molecular logic of WOX5..." represents a comprehensive effort to characterize the targets of WOX5 in the QC at the transcriptional and chromatin level. The ms would provide a valuable resource on a transcription factor that is the focus of a great deal of attention due its highly localized expression in the QC, the center of the root stem cell niche. The assays chosen are appropriate. However, the results and conclusions have several anomalies that are not clearly addressed in the ms right now. My view is that either more data needs to be collected or some further evaluation or discussion of the results needs to be made.

1. The authors state that WOX5 is a key regulator of the QC (line 52). But the results of the authors' transcriptional analysis leads them to conclude that WOX5 did not contribute substantially to DEGs between QC and columella. How can WOX5 be a key regulator of the QC and not control the differences between QC and columella? The results contradict several papers cited in the introduction, which the authors do not attempt to refute. This makes the primary conclusions unclear.

2. The authors cite previous work that showed non-cell autonomous effects for WOX5. Do the results suggest that non-cell autonomous regulation is the primary mode of WOX5's activity on gene regulation? In that case, it would seem that the authors should profile columella cells in the *wox5* mutant.

3. Alternatively, do the authors think that the variability in the profiles that they cite are the reason why they do not detect a substantial contribution of WOX5 to QC identity? i.e, were WOX5 targets missed in the WT tissue profiles or the mutant profile?

4. Given all the anomalies and low significance of the known differences between QC and columella, I think the authors need some in situ hybridization validation of the profiles. I realize that QC and columella are difficult tissues to probe, but perhaps some of the newer, sensitive protocols for in situs will help, e.g, HCR probes.

5. The paradoxical set WOX5-activated direct and indirect targets that are higher in the columella or mature tissues than QC is also puzzling. Do these genes also exhibit the bivalent pattern for chromatin modification?

6. Is it possible that the paradoxical set of genes is an indirect effect of the mutant? The assay for direct activation was done with 35S::WOX5, so presumably there could be many false positives among the direct activation set. This is a difficult aspect of the ms to validate due to the low expression level of the paradoxical set of genes in the QC. But I do think some kind of corroboration is needed to make this conclusion.

7. The analysis and how the different data types generated are difficult to follow. I think there needs to be more attention to the writing. There are some confusing parts that are cause sometimes by minor wording errors (see below) but often by just the explanation of the analysis and the lack of clarity of the conclusions in a given section.

Line 49: maize is also regarded to have a closed meristem. <https://academic.oup.com/aob/article/48/6/761/112964>

Line 50: take over instead of overtake

Line 63: it is not clear what is meant by a proximal meristem

Line 245: the terminology switches to QC-up. I can't see where this is defined. Seemingly the same as activated, but it is confusing.

Line 359: reconciled instead of concealed?

Extended data figure 1. It is not clear what genetic background the graphs represent. Panel C is labeled, but does that apply to A and B? It's ambiguous.

Dear Editor,

We greatly appreciate your and the reviewers' comments, which helped us to improve the manuscript significantly. We have carefully addressed all points raised by the reviewers. Our responses are highlighted in blue color in both the response letter and the revised manuscript. We sincerely hope that you will find the revised manuscript a significant improvement. Thank you once again for your time and endeavors.

Sincerely,

Thomas Laux, in the name of all authors

Referee #1:

The manuscript entitled "Deciphering the molecular logic of WOX5 function in the root stem cell organizer" by Zhang et al. reported the mechanism of WOX5-mediated QC functions in the root tip stem cell niche. The authors used the fluorescent-activated nuclei sorting and revealed the genes/pathways regulated by WOX5 and the accompanied epigenetic landscapes. Based on these data, they found that WOX5 served as both transcription activator and repressor, and identified a new pathway related to nitrate uptake as a downstream of WOX5. Furthermore, the data supported the idea that QC might function as a stem cell reserve. Overall, the manuscript provides many interesting data and will surely improve our understanding of the QC role in the root tip stem cell niche. The manuscript is well organized. I have only two minor suggestions for the authors to improve the manuscript.

Specific points:

1) The "Bivalent" chromatin domains discovered in QC are interesting. Some of the data were organized in Figure 3 and Supplemental data. It will be interesting to show some specific cases with bivalent epigenetic modification loci in Figure 3 (or other places in the main text), and have a discussion on some specific genes related to stem cells/QC or other pathways.

Response: We agree and, after searching the literature, have provided information on two examples from our set of WOX5-increased bivalent chromatin genes that are involved in root regulation (lines 288-290):

“Examples include the *ATP-BINDING CASSETTE C4 (ABCC4)* and *CELLULOSE SYNTHASE A2 (CESA2)* genes, which play crucial roles in regulating root development (Uragami et al, 2024) (Persson et al, 2007).”

And to the discussion (lines 411-416):

“A phenotypic analysis of root development has been published only for a few examples in this group of genes. The *ABCC4* gene encodes an *Arabidopsis* cytokinin efflux transporter and is crucial in restricting root elongation by controlling the active cytokinin flow (Uragami *et al.*, 2024). The *CESA2* gene encodes a component of a primary cell wall cellulose synthase complex and is essential for maintaining the structural integrity required for root growth (Persson *et al.*, 2007).”

2) Why did the authors choose H3K4me3 and H3K9ac as the epigenetic markers for gene activation but not choose H3K36me3? In plants, H3K36me3 is typically associated with gene activation. The authors may have an explanation in the main text.

Response: We agree that H3K36me3 would be another excellent choice as a mark associated with gene activation. To the best of our knowledge, however, it appears to be more closely linked to transcription elongation in plants (Lam *et al.* 2022). To explain why we chose the marks analyzed, we added (lines 185-190):

“We chose H3K9ac and H3K4me3 as marks associated with transcriptionally competent chromatin because previous studies showed that H3K9ac is directly modified by WOX5 in the *CDF4* gene (Pi *et al.*, 2015), and, together with H3K4me3, is often found near the TSSs where also WOX5 binds. As a mark for repressive chromatin, we chose H3K27me3, a hallmark of transcriptional downregulation in plants and animals (Wiles & Selker, 2017).”

Referee #2:

Major comments:

1. For the GO enrichments, were these performed against the TAIR10 genome or against only the detected transcripts from each experiment? Please clarify how the GO analyses were performed in the methods section (lines 544-545).

Response: We apologize that this was not laid out very well and have clarified this in the methods section (lines 684-685):

“The GO enrichment analyses were performed with the PlantGSEA database (Yi *et al.*, 2013), and redundant terms were removed using Revigo with default parameters (Supek *et al.*, 2011).”

2. All sequence data should be deposited at a public repository with accession numbers.

Response: We added the detailed information in the Data Availability section (lines 696-702):

“The ChIP-Chip data generated in this study have been deposited in the GEO database under

accession code GSE270459 (<https://www.ncbi.nlm.nih.gov/geo/query/acc.cgi?acc=GSE270459>, password: qhgdqeaevlgtzmp). The RNA-Seq, ATAC-Seq, and histone CUT&RUN data generated in this study have been deposited in the BioProject under BioProject ID PRJNA1127817 (<https://dataview.ncbi.nlm.nih.gov/object/PRJNA1127817?reviewer=q29edsb56q8j2loep7rt6dshoa>) and PRJNA1132792 (<https://dataview.ncbi.nlm.nih.gov/object/PRJNA1132792?reviewer=cf8e9ddjutkid9mtt7s9tlf7bg>).”

3. Additional discussion on which "cell cycle" genes are bound and regulated by WOX5 would be useful to tie into the introduction, which discusses the concept of mitotically active/inactive cell states. For example, are there particular classes of CYC genes (or other cell cycle regulators) that are bound and regulated by WOX5?

Response: We thank you for this suggestion. Indeed, we identified several additional cell cycle genes regulated by WOX5. We added these genes to the result part (lines 232-235):

“Importantly, the WOX5-regulated DEGs confirm and expand on known functions of WOX5 in the QC: repression of cell division (*CYCD3.3*, *KRP1*, *KRP2*, *SIM*, *BORI2*) and differentiation (*CDF4*, *BBM*, *RGF8*, *HB-8*, *QQS*), and promoting auxin signaling (*HAN*, *TAA1*) (Table EV23).”

And a detailed discussion in the discussion section (lines 431-439)

“In this regard, our data significantly expands previous knowledge of how these features are regulated. For example, previous studies showed that WOX5 promotes QC quiescence by directly repressing the D-type cyclin *CYCD3;3* (Forzani *et al*, 2014). Here, we show that WOX5 additionally activates the expression of the cyclin-dependent kinase inhibitor genes *KIP-RELATED PROTEIN 1* and 2, encoding negative regulators of cell division (Verkest *et al*, 2005), and the repressor of endomitosis *SIAMESE (SIM)* (Churchman *et al*, 2006). Moreover, we found that WOX5 represses the *BOREALIN RELATED INTERACTOR 2 (BORI2)* gene, which facilitates the segregation of chromosomes during cell division (Komaki *et al*, 2022). Thus, WOX5 promotes QC mitotic quiescence through multiple cell-cycle-related genes.”

Minor comments:

1. Line 27: replace "basic" with "basal".

Response: Thank you for spotting this. We changed it as suggested (line 25).

2. Line 28-29: suggest re-wording to "These data support a model for QC cells as both reserve stem cells and primed cells for prospective progenitor fates".

Response: Yes, this is much better. We changed it as suggested (lines 26-27).

3. Line 58-59 needs further elaborated/clarified. A study from Clark et al., 2019 Nature Comm (<https://doi.org/10.1038/s41467-019-13132-2>) characterized stem cell ubiquitous gene expression in Arabidopsis roots and therefore may be a better resource for identifying QC-enriched/specific genes rather than enhancer trap lines that are not mapped.

Response: Thank you for pointing this out. We have added this (lines 86-87):

“A recent study of the *Arabidopsis* root stem cell niche revealed stem-cell-specific gene networks (Clark *et al*, 2019), providing a resource for future studies.”

4. Line 71: suggest re-wording to "WOX5 expression is also positively regulated by auxin, suggesting a feed forward loop between WOX5 and auxin in QC cells".

Response: Thank you for pointing this out. We have changed this sentence as suggested (lines 77-78).

5. Line 72: suggest changing "dependent and independent" to "coordinated" and "parallel".

Response: We agree that this is much clearer and have changed this sentence as suggested (line 79).

6. Line 76: define EAR acronym.

Response: Thank you for spotting this omission. We added (line 94):

“.....ethylene-responsive element binding factor-associated amphiphilic repression (EAR) domain.....”

7. Line 146: suggest re-wording to "We also identified enriched GO biological processes in QC cells that have not previously been linked with root stem cells."

Response: We changed this as suggested (lines 170-171).

8. Line 173: was this observed "slightly higher" frequency of genes enriched for both chromatin marks more than random chance? A statistical test for this enrichment could help support this result.

Response: We apologize for this oversight. We have added the statistical test to Appendix Figure S2E.

9. Line 185: replace "Our" with "The".

Response: We agree and changed it as suggested (line 217).

10. Lines 204-205: For clarity, perhaps re-word to indicate that these genes are both bound and regulated by WOX5.

Response: We apologize that this was not clear and changed this sentence as suggested (lines 262-266):

“Therefore, we focused on these 812 upregulated and 259 downregulated DEGs, which are bound and regulated by WOX5 as the putative direct WOX5 targets for further analysis (**Fig. 3C**). We confirmed the two published directly repressed WOX5 target genes *CYCD3.3* and *CDF4* among the potential directly downregulated WOX5 targets (**Table EV24, 25**).”

11. Lines 247 and 255: GO terms are notorious for being incorrect sometimes or based on limited understanding of particular gene functions.

Response: After performing the GO analysis, we routinely checked whether an important GO term includes genes with published functions that confirm association with that GO term. We fully agree with this reviewer that this is important to be pointed out and added an example to the text (lines 315-322):

“To mitigate the limitations of GO term annotations, we reviewed published studies to address whether genes assigned to a given term had a confirmed corresponding function. Two such examples of the terms “epidermal cell differentiation” and “developmental maturation” are *EXPANSIN A7 (EXPA7)*, which is specifically required for root hair elongation (Lin *et al*, 2011) and *ATP-BINDING CASSETTE B4 (ABCB4)*, which regulates free IAA levels in the differentiation/maturation zone of the root, crucial for maintaining an auxin, root differentiation, and growth (Kubes *et al*, 2012).”

12. Line 254: change "basic" to "basal".

Response: Thank you for spotting this. Done as suggested (line 325).

Referee #3:

The manuscript by Zhang et al. "Deciphering the molecular logic of WOX5..." represents a comprehensive effort to characterize the targets of WOX5 in the QC at the transcriptional and chromatin level. The ms would provide a valuable resource on a transcription factor that is the focus of a great deal of attention due its highly localized expression in the QC, the center of the root stem cell niche. The assays chosen are appropriate. However, the results and conclusions have several anomalies that are not clearly addressed in the ms right now. My view is that either more data needs to be collected or some further evaluation or discussion of the results needs to be made.

1. The authors state that WOX5 is a key regulator of the QC (line 52). But the results of the authors' transcriptional analysis leads them to conclude that WOX5 did not contribute substantially to DEGs between QC and columella. How can WOX5 be a key regulator of the QC and not control the differences between QC and columella? The results contradict several papers cited in the introduction, which the authors do not attempt to refute. This makes the primary conclusions unclear.

Response: Thank you for pointing out this very important point. We apologize for making it not clear that a key finding of our study is that the main role of WOX5 appears to be regulating specific QC features (cell cycle, differentiation, auxin signaling) in a broader way than previously known together with novel functions, but does not to regulate QC-identity in the sense of the global transcriptomic differences between QC and CCs nuclei. We think that this is one interesting finding of our study. To make this clearer, we rephrased our statements in the manuscript to emphasize well-defined QC functions rather than a vaguely defined QC identity:

First, we rephrase the sentence that WOX5 is a key regulator of the QC to clarify what we mean (line 64-66):

"The *WUSCHEL RELATED HOMEODOMAIN 5 (WOX5)*, which is specifically expressed in the QC cells (Haecker *et al*, 2004; Sarkar *et al*, 2007) regulates several important features of the QC."

In the result section (lines 232-243):

“Importantly, the WOX5-regulated DEGs confirm and expand on known functions of WOX5 in regulating key features of the QC: repression of cell division (*CYCD3.3*, *KRP1*, *KRP2*, *SIM*, *BOR12*) and differentiation (*CDF4*, *BBM*, *RGF8*, *HB-8*, *QQS*), and promoting auxin signaling (*HAN*, *TAA1*) (Table EV23). On the other hand, comparing the WOX5-regulated genes with our nuclear QC and CC transcriptome data, we found only limited, though statistically significant, overlap (Appendix Fig. S4), including the upregulation by WOX5 of the published QC-specifically expressed genes *BABY BOOM* (*BBM*) and *ROOT MERISTEM GROWTH FACTOR 8* (*RGF8*) (Aida *et al.*, 2004; Denyer *et al.*, 2019; Fernandez *et al.*, 2013), and the starch accumulation repressor gene *Qua-Quine Starch* (*QQS*) (Li *et al.*, 2009), consistent with the absence of starch in the QC (Table EV23). Thus, WOX5 regulates multiple aspects of the key features of the QC, mitotic quiescence, differentiation state, and auxin signaling, rather than global transcriptome differences between QC and CC.”

And in the discussion (lines 428-447):

“How does WOX5 regulate QC function? Our data reveal that WOX5 has a lesser effect on global transcriptome differences between QC and CC cells but regulates specific key features of the QC, repression of cell divisions, and differentiation and promotion of auxin signaling. In this regard, our data significantly expands previous knowledge of how these features are regulated. For example, previous studies showed that WOX5 promotes QC quiescence by directly repressing the D-type cyclin *CYCD3;3* (Forzani *et al.*, 2014). Here, we show that WOX5 additionally activates expression of the cyclin-dependent kinase inhibitor genes *KIP-RELATED PROTEIN 1* and *2*, encoding negative regulators of cell division (Verkest *et al.*, 2005), and the repressor of endomitosis *SIAMESE* (*SIM*) (Churchman *et al.*, 2006). Moreover, we found that WOX5 represses the *BOREALIN RELATED INTERACTOR 2* (*BOR12*) gene, which facilitates the segregation of chromosomes during cell division (Komaki *et al.*, 2022). Thus, WOX5 promotes QC mitotic quiescence through multiple cell-cycle-related genes. Likewise, we found that in addition to the published repression of the differentiation gene *CDF4*, WOX5 also represses *HB-8*, which acts as a differentiation-promoting transcription factor of the vascular meristem (Baima *et al.*, 2001). Moreover, our results showed that WOX5 also activates *BBM* and *RGF8*, which play significant roles in the regulation of root stem cell maintenance (Aida *et al.*, 2004; Fernandez *et al.*, 2015), and *QQS*, which represses starch accumulation (Li *et al.*, 2009). In conclusion, the genomic analysis of WOX5 function in the QC revealed that WOX5 directly regulates multiple components of mitotic quiescence and differentiation inhibition.”

2. The authors cite previous work that showed non-cell autonomous effects for WOX5. Do the results suggest that non-cell autonomous regulation is the primary mode of WOX5's activity on gene regulation? In that case, it would seem that the authors should profile columella cells in the *wox5* mutant.

Response: If this reviewer refers to the published non-cell-autonomous function of WOX5 on the single layer of columella stem cells (CSCs), we agree that addressing the target genes of the non-cell-autonomous WOX5 function at a genomic scale would be fantastic. Still, it is currently impossible due to the lack of a robust fluorescent marker for CSCs. Therefore, this paper aimed to characterize the role of WOX5 in the stem cell organizer.

3. Alternatively, do the authors think that the variability in the profiles that they cite are the reason why they do not detect a substantial contribution of WOX5 to QC identity? i.e, were WOX5 targets missed in the WT tissue profiles or the mutant profile?

Response: We agree that omic data are notorious for some variability. However, we do not have any evidence that this would result in under-detecting a specific group of genes, e.g., QC-specific genes. We agree that it is important to address this possible limitation, which we did as follows (lines 375-380):

“Importantly, we could confirm the majority of DEGs, suggesting that the false positive rate of our approach is low. On the other side, however, we also found evidence that statistical variations between deep sequencing replicates can hamper the recognition of some WOX5-regulated DEGs. Therefore, while the identified DEGs have high confidence levels, we cannot exclude that some WOX5-regulated genes identified by our criteria may have been missed.”

4. Given all the anomalies and low significance of the known differences between QC and columella, I think the authors need some *in situ* hybridization validation of the profiles. I realize that QC and columella are difficult tissues to probe, but perhaps some of the newer, sensitive protocols for *in situ* will help, e.g, HCR probes.

Response: Please see also our responses to the previous queries. As this reviewer correctly pointed out, *in situ* hybridization in *Arabidopsis* roots often does not result in robust data, especially for weakly expressed genes. Therefore, we validated our results by (1) comparison

with published QC and columella reporter genes (**Fig. EV1A**), (2) RT-qPCR (**Table EV4**), (3) and by comparison with published QC, columella and other transcriptome data (**Fig. EV1C, D, Fig. 4E**). We generally can validate between 80 and 90% of the genomic data at a single gene level, suggesting a low false discovery rate.

5. The paradoxical set WOX5-activated direct and indirect targets that are higher in the columella or mature tissues than QC is also puzzling. Do these genes also exhibit the bivalent pattern for chromatin modification?

Response: We agree that this is a very appealing concept. According to this reviewer's suggestion, we scrutinized the paradoxical genes for WOX5-increased bivalent histone modification patterns. However, there is no significant overlap between these genes and WOX5-regulated bivalent marks.

6. Is it possible that the paradoxical set of genes is an indirect effect of the mutant? The assay for direct activation was done with 35S::WOX5, so presumably there could be many false positives among the direct activation set. This is a difficult aspect of the ms to validate due to the low expression level of the paradoxical set of genes in the QC. But I do think some kind of corroboration is needed to make this conclusion.

Response: We agree with this reviewer's notion. Due to the scarcity of QC cells, we have used ectopic stem cell induction by 35S::WOX5 as a proxy. We took the following measures to reduce false positives to the best of our possibilities.

We used a more stringent parameter than the standard setting (distCutOff = 600, minProbesInRow = 3; (Toedling *et al*, 2007)) to identify ChIP-enriched regions to reduce false positives (lines 565-566):

“This study used a high stringency parameter “thresholds = 2, distCutOff = 250, minProbesInRow = 5” to identify ChIP-enriched regions.”

We crossed the ChIP data set with the WOX5 transcript data to exclude the false positive further (**Fig. 3C**).

Furthermore, several observations are encouraging. Among the candidate direct WOX5 target genes selected by this procedure, we confirmed the only known published direct WOX5-target *CDF4* and *CYCD3.3*, indicating the reliability of our data. Furthermore, for three out of

four selected direct candidate genes that also show WOX5-regulated histone marks and chromatin accessibility (Fig. 5, Table EV36), including the member of the "paradoxical" subset *CEPR2*, we confirmed that they are functionally relevant by partial complementation of the *wox5-1* mutant.

Nevertheless, while these examples are encouraging, we agree that we cannot rule out the possibility that some of these candidate direct target genes are indirectly regulated. Further refinement involving a cell-specific functional analysis of appropriate mutants is a challenging task for the future and beyond the scope of this paper.

7. The analysis and how the different data types generated are difficult to follow. I think there needs to be more attention to the writing. There are some confusing parts that are caused sometimes by minor wording errors (see below) but often by just the explanation of the analysis and the lack of clarity of the conclusions in a given section.

Response: We apologize for the lack of clarity. We carefully reviewed the manuscript and rewrote substantial parts to clarify the data types. In addition to the clarifications mentioned in the responses to the previous queries, we have added more information for analysis and different data types,

lines 589-590:

“For RNA-Seq, comparisons were made between wild-type CCs and wild-type QC, and between *wox5-1* and wild-type QC.”

And lines 623-624:

“For CC and QC CUT&RUN, wild-type CCs and wild-type QC were compared. For *wox5-1* and wild-type QC CUT&RUN, *wox5-1* and wild-type QC were compared.”

And lines 654-655:

“For ATAC-Seq, comparisons were made between wild-type CCs and wild-type QC, and between *wox5-1* and wild-type QC.”

Line 49: maize is also regarded to have a closed meristem.

<https://academic.oup.com/aob/article/48/6/761/112964>

Response: Thank you for pointing out this mistake. We corrected this to common sunflower (line 53).

Line 50: take over instead of overtake

Response: Thank you for spotting this. We changed it as suggested (line 54).

Line 63: it is not clear what is meant by a proximal meristem

Response: We apologize for being unclear. We now clarified this in lines 42-45:

“The proximal stem cells give rise to the stele, endodermis, and cortex, the lateral ones to the epidermis and the lateral root cap, and the distal stem cells to the gravity-sensing columella (Dolan *et al.*, 1993).”

And rephrased the abovementioned sentence to (lines 56-58):

“Unlike the stem cells for all other root cell files, the asymmetric divisions of the CSCs give rise to daughter cells that do not partake in further divisions but undergo direct differentiation into columella cells (CCs) (Dolan *et al.*, 1993).”

Line 245: the terminology switches to QC-up. I can't see where this is defined. Seemingly the same as activated, but it is confusing.

Response: We are sorry for the confusion. We have now better defined the QC-up and CC-up as terms to describe the DEGs from comparing QC and CC nuclei in Lines 130-133:

“We identified 1709 differentially expressed genes (DEGs) expressed at a higher level in the QC nuclei (hereafter: QC-up DEGs) than in the CC nuclei. On the other hand, 2168 genes were expressed at higher levels in the CCs (hereafter: CC-up DEGs) than in QC (P-adj < 0.05, FC \pm 1.5) (Fig. 2A, Table EV2, 3).”

When referring to WOX5-regulated genes, we use the terms WOX5-activated and WOX5-repressed (lines 211-215):

“To identify WOX5-regulated genes in the QC, we compared the transcriptomes of sorted nuclei expressing *pWOX5:H2B-tdTomato* from *wox5-1* and wild-type roots (Fig. 1A-D, Table EV1). We found that 1423 genes were positively (“WOX5-activated”, wild-type QC > *wox5-1* QC) and 947 genes were negatively (“WOX5-repressed”, wild-type QC < *wox5-1* QC) associated with WOX5 activity (P-adj < 0.05; Fig. 3A, Table EV20, 21).”

Line 359: reconciled instead of concealed?

Response: Thank you for pointing this out. We have changed as suggested (line 470).

Extended data figure 1. It is not clear what genetic background the graphs represent. Panel C is labeled, but does that apply to A and B? It's ambiguous.

Response: As suggested, we have clarified the genetic backgrounds for all panels in the legend of Appendix Figure S1.

Dear Thomas,

Thank you for submitting a revised version of your manuscript. I sincerely apologise for the protracted assessment process due to delays in referee report submission.

We have now received input from two of the original reviewers, who now find that their previous concerns have been addressed satisfactorily and broadly recommend acceptance of the manuscript. Therefore, there now remain only a few editorial points that need addressing before I can extend official acceptance of the manuscript:

1. Please submit up to five keywords.
2. Please check that the funding information is correct and identical both in the manuscript and our online system. Currently, CIBSS Launchpad Programme funding is missing in our online system.
3. CRedit has replaced the traditional author contributions section because it offers a systematic, machine-readable author contributions format that allows for more effective research assessment. Please remove the Authors Contributions from the manuscript and use the free text boxes beneath each contributing author's name in our online submission system to add specific details on the author's contribution. More information is available in our guide to authors.
4. Please rename "Competing interests" section into "Disclosure and competing interests statement" (further info: <https://www.embopress.org/page/journal/14602075/authorguide#conflictsofinterest>).
5. Please rename tables EV1, 2, 3, 6-21, 24, 26-35 into "Dataset EV1", etc. Please adjust the numbering for the remaining EV Tables to "Table EV1", etc. Please upload datasets and tables as individual files, one for each table or dataset.
6. Please remove the Reagents and Tools table from the manuscript text file, as we require only the individually uploaded file that you have kindly provided.
7. Our data editors have flagged the following issues in figure legends that need correcting:
 - Please provide the exact p values in the legends of figures 2f; 5e, l; EV 1c; EV2a-c; EV 3b.
 - Please indicate the statistical test used for data analysis in the legends of figures 2a; 3a.
 - Please define the box plot in terms of minima, maxima, centre, bounds of box and whiskers, and percentile in the legend of figure 5e.
 - Please provide information on the nature and number of replicates in the legends of figures 2a; 3a.
 - Please describe the nature of replicates (e.g., biological or technical) in the legends of figures 5e, l.
8. Papers published in The EMBO Journal are accompanied online by a 'Synopsis' to enhance discoverability of the manuscript. It consists of A) a short (1-2 sentences) summary of the findings and their significance, B) 3-4 bullet points highlighting key results and C) a synopsis image that is 550x300-600 pixels large (width x height, jpeg or png format). You can either show a model or key data in the synopsis image. Please note that the image size is rather small and that text needs to be readable at the final size. Please send us this information together with the revised manuscript.

With best wishes,

leva

leva Gailite, PhD
Senior Scientific Editor
The EMBO Journal
Meyerohofstrasse 1
D-69117 Heidelberg
Tel: +4962218891309
i.gailite@embojournal.org

We realize that it is difficult to revise to a specific deadline. In the interest of protecting the conceptual advance provided by the work, we recommend a revision within 3 months (24th Dec 2024). Please discuss the revision progress ahead of this time with the editor if you require more time to complete the revisions.

Referee #2:

All suggested edits and requested information have been included in the revised manuscript.

Referee #3:

The authors have addressed my primary concerns largely by clarifying how their analyses affected QC function. The difficulty in probing the specific tissues that are implicated in the ms make that risky. The authors have done some additional validation with known markers. At this point, that should provide corroboration of the results, with the appropriate caveats stated.

The authors addressed the remaining editorial issues.

Dear Thomas,

Thank you for addressing the final editorial points. I am now happy to inform you that your manuscript has been accepted for publication in the EMBO Journal.

Before we forward your manuscript to the publishers, I would like to suggest minor edits in the manuscript abstract and synopsis. I have also written a short blurb that will accompany the title of your manuscript on our website. Please take a look at the text below and in the attached text file and let me know if any further edits are needed.

Blurb:

Transcriptomic and epigenomic analyses of WOX5's regulatory landscape in the Arabidopsis quiescent center.

Synopsis:

WOX5 is a homeodomain transcription factor that regulates function of the quiescent center (QC) - the stem cell organizer in the Arabidopsis root. Here, analysis of the WOX5-dependent transcriptional and epigenetic landscapes in the QC uncovers potential novel WOX5 roles, including in nitrate transport and the priming of differentiated gene expression in plant stem cell regulation.

- WOX5 acts both as a transcriptional activator and repressor in the QC.

- WOX5 not only regulates expression of target genes linked to its established roles in stem cell signaling and mitotic repression, but also to some novel functions, such as nitrate transport.

- WOX5 promotes weak expression of genes that are typically expressed in differentiated root cells, suggesting a role in priming the QC transcriptome for root regeneration after injury.

If you have any questions, please do not hesitate to contact the Editorial Office. Thank you for this contribution to The EMBO Journal and congratulations on a nice study!

Best wishes,

leva

leva Gailite, PhD
Senior Scientific Editor
The EMBO Journal
Meyerohofstrasse 1
D-69117 Heidelberg
Tel: +4962218891309
i.gailite@embojournal.org
